# seNorge_2018, daily precipitation and temperature datasets over Norway

Cristian Lussana[1], Ole Einar Tveito[1], Andreas Dobler[1], and Ketil Tunheim[1]

[1]Norwegian Meteorological Institute, Oslo, Norway

**Correspondence:** Cristian Lussana (critianl@met.no)

**Abstract.** seNorge_2018 is a collection of observational gridded datasets over Norway for: daily total precipitation; daily mean, maximum and minimum temperatures. The time period covers 1957 to 2017, and the data are presented over a high-resolution terrain-following grid with 1 km spacing in both meridional and zonal directions. The seNorge family of observational gridded datasets developed at the Norwegian Meteorological Institute (MET Norway) has a twenty-year long history and seNorge_2018 is its newest member, the first providing daily minimum and maximum temperatures. seNorge datasets are used for a wide range of applications in climatology, hydrology and meteorology. The observational dataset is based on MET Norway's climate data, which has been integrated by the "European Climate Assessment and Dataset" database. Two distinct statistical interpolation methods have been developed, one for temperature and the other for precipitation. They are both based on a spatial scale-separation approach where, at first, the analysis (i.e., predictions) at larger spatial scales are estimated. Subsequently they are used to infer the small-scale details down to a spatial scale comparable to the local observation density. Mean, maximum and minimum temperatures are interpolated separately, then physical consistency among them is enforced. For precipitation, in addition to observational data, the spatial interpolation makes use of information provided by a climate model. The analysis evaluation is based on cross-validation statistics and comparison with a previous seNorge version. The analysis quality is presented as a function of the local station density. We show that the occurrence of large errors in the analyses decays at an exponential rate with the increase in the station density. Temperature analyses over most of the domain are generally not affected by significant biases. However, during wintertime in data-sparse regions the analyzed minimum temperatures do have a bias between $2°C$ and $3°C$. Minimum temperatures are more challenging to represent and large errors are more frequent than for maximum and mean temperatures. The precipitation analysis quality depends crucially on station density: the frequency of occurrence of large errors for intense precipitation is less than 5% in data-dense regions, while it is approximately 30% in data-sparse regions. The open-access datasets are available for public download at: daily total precipitation (DOI:https://doi.org/10.5281/zenodo.2082320 Lussana, 2018b) ; daily mean (DOI:https://doi.org/10.5281/zenodo.2023997 Lussana, 2018c) , maximum (DOI:https://doi.org/10.5281/zenodo.2559372 Lussana, 2018e) and minimum (DOI:https://doi.org/10.5281/zenodo.2559354 Lussana, 2018d) temperatures

# 1 Introduction

Long-term observational gridded datasets of near-surface meteorological variables are widely used products. In climatology, they have been used for example to monitor the regional climate (Simmons et al., 2017) and to validate and bias-correct climate simulations (Kotlarski et al., 2017). In meteorology, they are used at national meteorological institutes, such as the Norwegian
Meteorological Institute (MET Norway), to monitor and report the weather conditions. In hydrology, they are used as external forcing for hydrological and snow modeling (Saloranta, 2012; Skaugen and Onof, 2014; Magnusson et al., 2015).

seNorge_2018 is a collection of four long-term observational datasets over Norway covering the 61-year time period 1957-2017 for: daily total precipitation (RR), daily mean temperature (TG), daily minimum (TN) and maximum (TX) temperatures. It builds upon the previous work on establishing MET Norway's observational datasets (Tveito and Førland, 1999; Lussana
et al., 2018a, b) and the core of its statistical interpolation method is the Optimal Interpolation (OI, Gandin and Hardin, 1965; Kalnay, 2003). A review of the relevant literature for our spatial interpolation applications is given in the paper by Lussana et al. (2018a).

Like the previous versions of seNorge, precipitation and temperature data are provided on a high-resolution grid with 1 km grid spacing in both meridional and zonal directions. seNorge_2018 aims at achieving a higher effective resolution of the
15 analyzed (or predicted) fields than the previous versions. It is worth spending a few words on effective resolution in OI. The difference between grid spacing and resolution is described by Grasso (2000). In the context of numerical modeling, Walters (2000) defines the effective resolution as "the minimum wavelength the model can describe with some required level of accuracy (not defined)" and it concludes that as many as 10 gridpoints may be required to properly represent a field. As pointed out by Pielke (2001), there is a subjective component in the number of gridpoints needed to resolve a feature in a field. In contrast
to in-situ observations which represent point values, our gridded analyses produce areal averages. What this means is that for each grid point, we calculate weighted averages of the nearest observations. The larger the extensions of the spatial supports for these averages, the lower the effective resolution of the analysis fields. In short, it is the availability of measurements that determines the highest possible effective resolution, irrespective of the chosen grid spacing, with topographic complexity a compounding factor (Hofstra et al., 2008). The settings used in the interpolation must consider this limitation, and if the same
settings are to be used over the whole area, then the sub-region of lowest station density may dictate the effective resolution of the entire domain.

The following definitions of spatial scales are used in the text. Regional scale coincides with the whole domain. Given the importance of the observational network, at an arbitrary point we refer to scales that are defined with respect to the station distribution in its surroundings. Sub-regional scale (or local scale) defines an area -around the point- that includes dozens
of observations (10-100). Small-scale defines an area that includes few observations (1-10). Unresolved scale refers to those spatial scales that are smaller than the average distance between a station and its closest neighbours, such that atmospheric fields could not be properly represented by the observational network.

The main original aspect of our research is that the spatial interpolation methods automatically adapt OI settings to the local station density, such that in data-dense regions the spatial supports of the areal-averaged analyses are smaller than in data-

sparse regions. In other words, the effective resolution of the analysis fields is higher in data-dense than in data-sparse regions. Because the spatial analysis depends on station density, the Integral Data Influence (IDI: Uboldi et al., 2008; Lussana et al., 2010) has been used as a diagnostic parameter to quantify the effects of station density on the analysis.

The presented research includes several other original aspects. In the case of precipitation, the measurements have been adjusted for the wind-induced under-catch in a way that is consistent with the method proposed by Wolff et al. (2015). A multi-scale OI scheme has been implemented on precipitation relative anomalies with respect to a reference field that captures the field variability at unresolved spatial scales. The reference fields are the monthly totals derived from a regional climate simulation with a resolution of 2.5 km. The climate simulation is based on the dynamical downscaling of the global reanalysis ERAInterim and it is available for the time period 2003-2016. In the paper by Crespi et al. (2019), it has been demonstrated that the combination of the same model fields with observed data do improve the representation of monthly total precipitation over Norway. Masson and Frei (2014) proved that the use of a reference field as a first guess for the precipitation patterns is a successful approach also in the Alps. They found that daily precipitation over the Alpine region is well represented by using the seasonal precipitation mean as a single predictor field in Kriging with external drift.

In the case of temperature, seNorge_2018 is the first seNorge dataset that includes daily minimum and maximum temperatures. The availability of these two additional variables allow for the computation of several more indices for climate variability and extremes, such as the ones reported in the paper by Zhang et al. (2011). The three temperature variables are treated separately with the same interpolation method. With respect to seNorge2 (Lussana et al., 2018b), the regional spatial trend of temperature is obtained as the blending of a much larger number of sub-regional trends. The analysis method has been implemented on a gridpoint-by-gridpoint basis in order to take advantage of a local Kalman gain.

The structure of the paper is as follows. Section 2 presents the observational network and the regional climate simulation used as the precipitation reference. Furthermore, IDI is described and discussed in Sec. 2 as for spatial analysis we regard this parameter as one of the basic properties characterizing a station, such as e.g. its altitude or the geographical location. The spatial interpolation methods are described in Sec. 3. An example application for precipitation is presented in Sec. 4. The features of seNorge_2018 daily temperature fields are very much similar to those displayed in e.g. Figs.(4)-(6) in the paper by Lussana et al. (2018b), since the grid is the same and the spatial analyses are based on the same principles. For this reason, example applications for temperature are not included. Section 5 presents the validation of seNorge_2018, that is largely based on cross-validation (CV) and comparison against seNorge2. Then, the results are discussed in Sec. 6.

## 2 Data

### 2.1 Observations

The in-situ observations are retrieved from MET Norway's climate database and the European Climate Assessment and Dataset (ECA&D, Klein Tank et al., 2002). The spatial domain covers the Norwegian mainland, plus an adjacent strip of land extending into Sweden, Finland and Russia in order to reduce boundary effects along the Norwegian border. The observations have been quality controlled by experienced staff and with the help of automatic procedures, such as the spatial consistency test described

by Lussana et al. (2010). The variables are defined as following: TG is the 24-hour average between 06:00 UTC of the day, reported as time-stamp and 06:00 UTC of the previous day; RR is the accumulated precipitation over the same time interval as TG, moreover RR data has been corrected for the wind-induced under-catch of the gauges; TX and TN are, respectively, the maximum and minimum observed temperatures between 18:00 UTC of the day reported as time-stamp and 18:00 UTC of the previous day. TG and RR share the same day-definition so as to serve hydrological applications (Saloranta, 2016; Skaugen and Mengistu, 2016). As a result of choices made in the past at MET Norway, TX and TN have a different day definition than RR and TG.

The measured RR value (i.e., $RR_{raw}$) at an arbitrary location is adjusted for wind-induced under-catch of solid precipitation by means of a procedure similar to the one presented by Wolff et al. (2015):

$$\alpha = \tau_1 + (\tau_2 - \tau_1)\left\{\exp\left[(\text{TG} - T_\tau)/s_\tau\right]/(1 + \exp\left[(\text{TG} - T_\tau)/s_\tau\right])\right\} \tag{1}$$

$$\gamma = [1 - \alpha]\exp\left[-(W/\theta)^\beta\right] + \alpha \tag{2}$$

$$\text{RR} = \gamma^{-1}\,\text{RR}_{raw} \tag{3}$$

where TG is extracted from the analysis field (Sec. 3.1) so as to always have a temperature estimate; W is the ten-metre wind speed at the station location extracted from a gridded dataset derived from numerical model output. The (NORA10, Reistad et al., 2011) wind speed dataset, which covers the whole time period 1957-2017, has been downscaled onto the 1 km grid by using a quantile mapping approach (Bremnes, 2004) to match the climatology of the high-resolution numerical weather prediction model (AROME-MEtCoOp, Müller et al., 2017). The wind dataset is available for public download at http://thredds.met.no/thredds/catalog/metusers/klinogrid/KliNoGrid_16.12/FFMRR-Nor/catalog.html. In the original paper by Wolff et al. (2015), they were considering sub-daily precipitation measurements and both temperature and wind were measured at the same location as the precipitation. We are operating under different conditions and the requirement of having temperature and wind measurements together with precipitation would reduce the number of suitable observations to a very small subset. As a consequence, in Eqs. (1)- (2) we had to use parameter values which are different from those used by Wolff et al. (2015). We have decided to use seNorge version 1.1 (Mohr, 2008, 2009) as a reference for the extreme values returned by the precipitation adjustment. seNorge version 1.1 includes a precipitation correction based on geographical parameters, summarized in site exposure classes such that a systematic increase of precipitation is carried out. The correction presented in Eqs. (1)- (3) takes advantage of wind and temperature estimates but we do not expect the extreme values of those two corrections to differ significantly. The parameter values used in Eqs. (1)- (3), which have been optimized to better match seNorge version 1.1 extremes, are: $\theta = 4.7449$, $\beta = 0.6667$, $\tau_1 = 0.4930$, $\tau_2 = 0.9134$, $T_\tau = 0.9134$, $s_\tau = 0.7759$.

Figure 1 shows the observational network and its evolution in time. The number of available observations was rather stable from 1957 to 2000. In the following decade, the number of RR observations dropped to 500, which was the minimum value, and then it gradually increased again to over 600 in recent years. The number of temperature observations has been constantly increasing since year 2000, and for 2017 there are about twice as many stations as in 1957. The meteorological stations have been mainly installed to monitor the weather in cities and villages, so the network is denser in urban areas. In the mountainous regions, the digital elevation model (resolution of 1 km$^2$) can reach 2000 m but most of the stations are located below the

elevation of 1000 m. A difference in the station density between the southern and the northern portions of the domain is also clearly visible, with a higher density in the south of Norway. Ideally, spatial interpolation would require a denser network of observations where the variance of the field is larger, in order to get a fine-scale representation of the field where it varies the most. However, this is hardly the case in most situations because of the inherent difficulties in station installation and maintenance over complex terrain and in remote areas. As a result, we should expect better performances of the interpolation methods over urban areas and larger analysis uncertainties over data-sparse areas, such as mountainous regions.

## 2.2 Reference fields for spatial interpolation of precipitation

The reference fields are derived from long-term averages calculated from the output of a high-resolution numerical model. The reference datasets used for precipitation are based on hourly precipitation provided by the climate model version of HARMONIE (version cy38h1.2), a seamless NWP model framework developed and used by several national meteorological services. HARMONIE includes a set of different physics packages adapted for different horizontal resolutions. For the high-resolution, convection permitting simulations in this case, the model has been set-up with AROME physics (Seity et al., 2011) and the SURFEX surface scheme (Masson et al., 2013). The climate runs have been carried out within the HARMONIE script system, covering the period July 2003 to December 2016 on a 2.5 km grid over the Norwegian mainland. More details on the climate model can be found in Lind et al. (2016), references therein and on https://www.hirlam.org/trac/wiki/HarmonieClimate. The numerical model does not include measurements from the network of rain-gauges. The mean monthly total precipitation fields have been computed considering the available hourly data and they have been used as reference fields for the spatial interpolation of precipitation as described in Sec. 3.2.

Over our domain, we have chosen not to use precipitation climatologies derived by observational gridded datasets as the reference because in some regions the observational network is extremely sparse (Fig. 1).

## 2.3 Integral Data Influence

IDI is similar to the degrees of freedom introduced by Cardinali et al. (2004) and it has been used also to evaluate the distribution of weather stations (Horel and Dong, 2010). In practice, IDI is obtained as the result of an OI performed by arbitrarily assigning a value of 1 to the observations (i.e., maximum amount of available information) and the reference value of 0 to the background (i.e, basic amount of information available everywhere). The analytical function that usually represents the background error correlation in OI, in the case of IDI is representing the station influence on the analysis according to a predefined metric. This metric is defined as a function of the geographical parameters. For an arbitrary point in space, the geographical parameters are stored in a vector $\mathbf{r}$ having four components: latitude, longitude, altitude and land area fraction (i.e. fraction of land in the 1 km square box centered at the point). The land area fraction is introduced here and used in Sec. 3.1. Functions are applied to pair of points, such as: $d(\mathbf{r},\mathbf{s})$ returns the horizontal (radial) distance in km between $\mathbf{r}$ and $\mathbf{s}$; $z(\mathbf{r},\mathbf{s})$ returns their absolute elevation difference; $w(\mathbf{r},\mathbf{s})$ returns their absolute land area fraction difference. The correlation function between two points $\mathbf{r}$ and $\mathbf{s}$ is

based on Gaussian functions of the form:

$$f_u\left(\mathbf{r},\mathbf{s};D\right) = \exp\left\{-\frac{1}{2}\left[\frac{u(\mathbf{r},\mathbf{s})}{D}\right]^2\right\} \tag{4}$$

where: $u()$ is an arbitrary function, such as the ones previously defined, applied to the points; $D$ is a reference length scale governing the decreasing rate. We have chosen to model the station influence using Gaussian functions. For TG, TX and TN, the station influence is factorized into the product of two Gaussian functions: one depending on distances, such that in Eq. (4) $u = d()$ and $D =$50 km; the other depending on elevation differences, with $u = z()$ and $D =$200 km. In the case of RR, the station influence depends only on distances, therefore $u = d()$ and $D =$10 km. The values of the de-correlation length scales used for temperature are consistent with the findings of Sec. 3.1. For precipitation, the value chosen is representative of the smallest spatial scales used in multi-scale OI of Sec. 3.2.

For the purpose of evaluation in Sec. 5, the CV-IDI at station locations (i.e., IDI at a station location computed without considering the presence of that station) is introduced to link the CV statistics to the IDI of the hypothetical gridpoint represented by a station location.

In the two maps of Fig. 2, the IDI is shown for TG and RR based on the station distributions shown in Fig. 1. The IDIs for TX and TN are very similar to TG. In the vicinity of an observation the IDI field is approximately equal to 1 whereas for data sparse areas its value is close to 0. The IDI and CV-IDI values have been arbitrarily divided into four classes: values smaller than 0.45 define observations/gridpoints in data-sparse regions (i.e., where the station influence on the analysis is very limited); values larger than 0.85 define observations/gridpoints in data-dense regions (i.e., where the station influence on the analysis is substantial), then two transition classes between data-dense and -sparse regions have been defined.

For temperature, elevation plays a predominant role and even only a few stations at higher elevations can provide a reasonable approximation of the sub-regional near-surface temperature lapse rate. Fig. 2 shows that the regions where our observational network is sparser are the Northernmost part of Norway (i.e., above latitude 69 N) and the Scandinavia mountains between latitude 66-68 N. For precipitation, we have decided to not consider elevation in the spatial analysis because we are aware that our network is very sparse at higher elevations (see Fig. 1).

For precipitation, the IDI map in Fig. 2 shows values larger than 0.85 for those regions where the observational network can reconstruct patterns in the analysis fields where the small-scales have a resolution of approximately 10 km. The largest continuous regions with IDI larger than 0.85 are located in the southern part of the domain (i.e, below latitude 65 N) and mostly along the coast.

Fig. 2a and Fig. 2b show the close relationships between CV-IDI and the station density. As shown by Fig. 2c, at station locations IDI has a smaller range of values than CV-IDI. In fact, even an isolated station constitutes more information than the background alone, while an isolated gridpoint must have IDI equal to 0 as it is CV-IDI at an isolated stations.

# 3 Spatial Interpolation methods

The notation used is based on both Ide et al. (1997) and Sakov and Bertino (2011). The number of gridpoints is $m$. The number of observations is $p$. Upper-case bold symbols are used for matrices, lower-case bold symbols for vectors and italic symbols for scalars. For an arbitrary matrix $\mathbf{X}$, $\mathbf{X}_j$ means the $j$th column; $\mathbf{X}_{i,:}$ the $i$th row; and $\mathbf{X}_{ij}$ the element at the $i$th row and $j$th column. For an arbitrary vector $\mathbf{x}$, $\mathbf{x}_i$ denotes the $i$th element. The superscripts on the upper left hand corner of a symbol identify: analysis $a$; background $b$; observation $o$. Upper accents have been used too. In the case of temperature, where we iterate over the gridpoints, the notation $\overset{i}{\mathbf{X}}$ indicates that matrix $\mathbf{X}$ is valid for the $i$th gridpoint and in this sense we may refer to it as a local matrix. In the case of precipitation, where we iterate over spatial scales, those length scales are indicated with greek letters and the notation $\overset{\alpha}{\mathbf{X}}$ indicates that matrix $\mathbf{X}$ is obtained as a function of the spatial scale of $\alpha$ km. Upper accents are not used only for matrices, for instance the in-situ observations are stored in the $p$-vector $\mathbf{y}^o$ but in the following we will refer to the $\overset{i}{p}$-vector $\overset{i}{\mathbf{y}}{}^o$ of the nearest observations to the $i$th gridpoint.

## 3.1 Statistical interpolation of temperature

The same interpolation scheme is used for the mean, the maximum and the minimum daily temperature. The physical consistency among the three variables is assured by post-processing the independently analyzed datasets and for each gridpoint we impose that: TN is always smaller or equal to TG; TX is always greater or equal to TG. The cross-checking is further discussed in Sec. 6.1.

The spatial interpolation is implemented on a gridpoint-by-gridpoint basis. It combines a regional pseudo-background field, that is the weighted average of numerous sub-regional fields, with the observations. The temperature analysis at the generic $i$th gridpoint is written as:

$$\mathbf{x}_i^a = \mathbf{x}_i^b + \overset{i}{\mathbf{K}}_{i,:} \left( \overset{i}{\mathbf{y}}{}^o - \overset{i}{\mathbf{y}}{}^b \right) \tag{5}$$

$\overset{i}{\mathbf{y}}{}^o$ and $\overset{i}{\mathbf{y}}{}^b$ are $\overset{i}{p}$-vectors of the nearest stations to the $i$th gridpoint.

The local Kalman gain in Eq. (5) is:

$$\overset{i}{\mathbf{K}}_{i,:} = \overset{i}{\mathbf{G}}_{i,:} \left( \overset{i}{\mathbf{S}} + \varepsilon^2 \overset{i}{\mathbf{I}} \right)^{-1} \tag{6}$$

$\overset{i}{\mathbf{I}}$ is the $\overset{i}{p} \mathrm{x} \overset{i}{p}$ identity matrix and $\varepsilon^2 \equiv \sigma_o^2/\sigma_b^2$ is the ratio between the constant observed ($\sigma_o^2$) and pseudo-background ($\sigma_b^2$) error variances that has been set to 0.5, as for seNorge2 (Lussana et al., 2018b). The local pseudo-background error correlation matrices are defined on the basis of the correlation function between pair of points $\rho^T(\mathbf{r}_j, \mathbf{r}_k)$ as:

$$\rho^T(\mathbf{r}_j, \mathbf{r}_k) = f_d(\mathbf{r}_j, \mathbf{r}_k; D_i^h) \ \ f_z(\mathbf{r}_j, \mathbf{r}_k; D^z) \ \ [1 - (1 - w_{min})|w(\mathbf{r}_j, \mathbf{r}_k)|] \tag{7}$$

such that the correlation between the $j$th gridpoint and the $k$th station is $\overset{i}{\mathbf{G}}_{jk} = \rho^T(\mathbf{r}_j, \mathbf{r}_k)$. Analogously, the correlation between the $j$th station and the $k$th station is $\overset{i}{\mathbf{S}}_{jk} = \rho^T(\mathbf{r}_j, \mathbf{r}_k)$. The Gaussian functions $f$ are defined in Eq. (4). A formulation

similar to Eq. (7) has been used in the paper by Lussana et al. (2009), in that case the land area fraction has been replaced by the land use. $w_{min}$ sets the minimum value for the factor related to land area fraction when $w(\mathbf{r}_i, \mathbf{r}_j)$ is maximum (i.e., equals to 1). $D^z$ and $w_{min}$ are fixed over the domain, while $D_i^h$ is allowed to vary between gridpoints, although with some restrictions. In an ideal situation of a very dense observational network, one may consider to rely on adaptive estimates for

the three parameters. This is not the case for our station distribution, so we have opted for a "hybrid" configuration (i.e., $D^z$ and $w_{min}$ fixed; $D^h$ adaptive) that would return robust estimates. The impact of large land area fraction differences on $\rho^T$ is less dramatic than those of large horizontal or elevation differences and it also impacts only a limited number of stations along the coast. Eventually, we have manually set $w_{min} = 0.5$ to achieve the desired effect of attenuating the influence of coastal areas over inland areas and vice versa, while at the same time avoiding the introduction of sharp gradients between those two

regions. The optimization procedure for $D_i^h$ and $D^z$ is described in the following of this section.

    The pseudo-background $x_i^b$ in Eq. (5) is the blending of $n$ sub-regional pseudo-backgrounds and it is in many ways similar to those described by Lussana et al. (2018b). Each sub-regional pseudo-background is defined by a centroid and it includes only the 30 stations closest to this centroid. The pseudo-background field with centroid at $\mathbf{r}_c$ is the $m$-vector $\overset{c}{\mathbf{x}}{}^b$ and its value at the $i$th gridpoint is $\overset{c}{\mathbf{x}}{}_i^b$. The seNorge_2018 domain has been divided on a 50x50 grid, each cell is a 24 km by 31 km rectangular box

and the nodes (i.e., centres of the cells) are the "candidate" centroids. If a node is inside the domain and has at least 30 stations in a neighbourhood of 250 km, then it is a suitable centroid. Those 30 temperature observations are used to estimate a sub-regional pseudo-background field as a function of the elevation only. The analytical function used to model the vertical profile of temperature is the one proposed by Frei (2014) for the alpine region and its parameters have been obtained by fitting the function to the aforementioned 30 observations. We assume that 30 observations can provide a reliable fitting. The generic $c$th

pseudo-background field $\overset{c}{\mathbf{x}}{}^b$ is derived directly from the digital elevation model by assuming that the $c$th sub-regional vertical temperature profile is valid for the whole domain. By using a 50x50 grid, the number of sub-regions $n$ is usually between 500 and 600 and there are significant overlaps between neighbouring sub-regions, such that the continuity of the regional pseudo-background is guaranteed. Finally, $\mathbf{x}_i^b$ is a weighted average of $n$ values:

$$\mathbf{x}_i^b = \frac{\sum_{c=1}^{n} \overset{c}{\mathbf{x}}{}_i^w \overset{c}{\mathbf{x}}{}_i^b}{\sum_{c=1}^{n} \overset{c}{\mathbf{x}}{}_i^w} \qquad (8)$$

where the weights at the $i$th gridpoint $\overset{c}{\mathbf{x}}{}_i^w$ are the $n$ IDI values (Sec. 2) and $\overset{c}{\mathbf{x}}{}_i^w$ is computed considering only those stations included in $c$th sub-regional pseudo-background field. The settings used in the IDI calculation are similar to those used for precipitation in Fig. 2, in the sense that the station influence decays with horizontal distance only and its de-correlation length scale is set to 27.5 km, that is the average of a box width and height on the 50x50 grid.

    The optimization of $D^z$ and $D_i^h$ of Eq. (7) is based on the statistics of the innovation (i.e. observation minus background) at

station locations. As described by Desroziers et al. (2005), the elements of the background error covariance matrix at station locations, which is modeled by us as $\sigma_b^2 \mathbf{S}$, should match the innovation sample covariances. In Tables 1-3, the values of the parameters determining $\sigma_b^2 \mathbf{S}$ are shown for a selection of year (1960, 1970, 1980, 1990, 2000, 2010) in the assumption of a constant $D^h$ (i.e., $D_i^h = Dh, \forall i = 1, \ldots, m$). Note that we have also added the average number of stations available for a

specific year, the average distances between them and the estimated observation error variance, which is not strictly required to compute $\mathbf{S}$ and it is set to be half of $\sigma_b^2$ in our analysis. The TN error variances are significantly higher than those for TG and TX, thus indicating that TN is a more challenging variable to interpolate. $D^h$ and $D^z$ do not differ significantly among TG, TX and TN, probably because the common observational network constitutes the major constraint in determining their value. This justifies our choice to set $D^z = 210$ m for the three variables. The parameter values in the Tables are more influenced by the majority of stations that are located in station-dense areas. Therefore, the value of $D^h = 55$ km can be considered as a suitable reference for the minimum allowed $D_i^h$ value. The procedure used for the $D_i^h$ estimates is similar to the one described for the regional pseudo-background field. $D_i^h$ is a weighted average as the one reported in Eq. (8) where $\overset{c}{\mathbf{x}}_i^b$ is replaced by the $c$th length scale, which is constant for all gridpoints. For the $c$th sub-region this length scale is set to the average distance between a station and its nearest 3 stations, provided that this distance is larger than $D^h = 55$ km, otherwise $D^h = 55$ km is used. In this way, the analysis in data-sparse regions is the result of an interaction between a few (approximately four) stations. At the same time, we take advantage of data-dense areas to increase locally the effective resolution of the analysis without destroying the continuity of the field. Note that the use of extremely different $D^h$ values between data-dense and sparse areas (i.e., with differences around one order of magnitude or more) would result in a rather confusing field to look at. In those cases it would be better to split the domain into sub-domains and operate independently on them.

## 3.2 Statistical interpolation of precipitation

The multi-scale OI analyses are the results of successive approximations of the observations over a sequence of decreasing spatial scales that at station locations converge to the observed values.

The interpolation scheme is not applied directly to the RR values (the vector of the raw observed values adjusted for the wind-induced under-catch is indicated as $\mathbf{y}^{\mathrm{rr}}$) but to their anomalies relative to a reference field of monthly precipitation (see Sec. 2, indicated with the abbreviation ref in the following). In addition, a Box-Cox transformation with power parameter set to 0.5 is used and the transformation is indicated with the function $g()$. A similar transformation has been suggested by Erdin et al. (2012), though in the context of combination of radar with gauge data. The $i$th element of $\mathbf{y}^o$ used in multi-scale OI is:

$$\mathbf{y}_i^o = g\left(\mathbf{y}_i^{\mathrm{rr}}/\mathbf{y}_i^{\mathrm{ref}}\right) \tag{9}$$

The analysis procedure can be written as:

$$\mathbf{x}^a = g^{-1}\left[\mathcal{M}_2 \circ \mathcal{M}_3 \circ \ldots \circ \mathcal{M}_\eta\left(\tilde{\mathbf{x}}^a\right)\right] \odot \mathbf{x}^{\mathrm{ref}} \tag{10}$$

where the three fundamental operations are: (1) the composition of several applications of the same statistical interpolation model down a hierarchy of spatial scales of $\{\eta\,\mathrm{km}, \ldots, 3\,\mathrm{km}, 2\,\mathrm{km}\}$, such that the results of a model application are used to initialize the successive one, $\circ$ stands for model composition and $\mathcal{M}_\eta$ stands for the application of the statistical model to the largest length scale of $\eta$ km, $\tilde{\mathbf{x}}^a$ is the average of the $\mathbf{y}^o$ elements; (2) the Box-Cox inverse-transformation $g^{-1}()$; (3) the elementwise multiplication of vectors $\odot$ to transform the relative anomalies into RR values and at the same time include the effects of unresolved spatial scales. Ideally, the sequence of spatial scales to be used in Eq. (10) should be bounded between

a very large scale (e.g, half the largest domain dimension) and a fine scale corresponding to the average distance between two stations in data-dense areas. The number of scales in between those two extremes is not critical for the final results, provided that they are enough to guarantee a continuous analysis field in all situations. For seNorge_2018, we are using approximately 100 scales with a minimum of 2 km and a maximum of 1400 km. According to Thunis and Bornstein (1996); Orlanski (1975), those spatial scales range from the regional synoptic down to the lower boundary of the mesoscale. The sequence is unevenly spaced as the difference between two consecutive scales is somewhat proportional to their values.

The step-by-step description of the model $\overset{\alpha}{\mathbf{x}}{}^a = \mathcal{M}_\alpha \left( \overset{\beta}{\mathbf{x}}{}^a \right)$ for two arbitrary consecutive length scales of $\beta$ km and $\alpha$ km (with $\beta > \alpha$) is:

$$\overset{\alpha}{\mathbf{x}}{}^b \quad = \quad \overset{\alpha}{\mathbf{\Psi}}\overset{\beta}{\mathbf{x}}{}^a \tag{11}$$

$$\overset{\alpha}{\mathbf{K}} \quad = \quad \overset{\alpha}{\mathbf{G}} \left( \overset{\alpha}{\mathbf{S}} + \varepsilon^2 \overset{\alpha}{\mathbf{I}} \right)^{-1} \tag{12}$$

$$\overset{\alpha}{\mathbf{x}}{}^a \quad = \quad \overset{\alpha}{\mathbf{x}}{}^b + \overset{\alpha}{\mathbf{K}} \left( \mathbf{y}^o - \overset{\alpha}{\mathbf{H}}\overset{\alpha}{\mathbf{x}}{}^b \right) \tag{13}$$

In order to reduce the multi-scale OI computational expenses, the original 1 km grid is aggregated onto a new coarser grid, with aggregation factor equal to the integer value nearest to $(\alpha/2)$ km. The aggregation groups several smaller rectangular boxes into a bigger one (e.g., if $\alpha = 8$ km then 16 of the 1 km by 1 km boxes are aggregated into a single box measuring 4 km by 4 km). The analysis at scale $\beta$ is used as the background for scale $\alpha$ in an OI scheme. The analysis values are transferred between the two non-matching grid by the operator $\overset{\alpha}{\mathbf{\Psi}}$ of Eq. (11), that is a bilinear interpolation mapping vectors on the (coarser) $\beta$-grid to vectors on the (finer) $\alpha$-grid. The observation operator $\overset{\alpha}{\mathbf{H}}$ of Eq. (13) is also a bilinear interpolation transforming vectors on the $\alpha$-grid to $p$-vectors. $\varepsilon^2$ is set 1, which means that observations and background are assumed the have the same error variances. The background error correlation matrices of Eq. (12) are defined on the basis of the correlation function $\rho^R$:

$$\rho^R \left( \mathbf{r}_j, \mathbf{r}_k \right) = f_d \left( \mathbf{r}_j, \mathbf{r}_k; \alpha \right) \tag{14}$$

The Gaussian function $f$ is defined in Eq. (4) and the notation is similar to Eq. (7). Note that the length scales enter multi-scale OI of Eq. (10) through the correlation function of Eq. (14).

An OI scheme such as the one presented in Eq. (11)- (14) is realizing a low-pass filter whose cut-off wavelength is approximately $\alpha$ km (Uboldi et al., 2008) so every iteration over a smaller spatial scale returns a field with more fine-scale details in it. For a given element of the observation vector, there may be a *critical* scale at which the background coincides exactly with the observed value such that its contribution to the innovation in Eq. (13) (i.e., the term in parentheses) is equal to zero. As a consequence, the analysis values $\overset{\alpha}{\mathbf{x}}{}^a$ in the surroundings of that observation are unlikely to change in the passage between scales $\alpha$ and $\beta$ and those analysis values are also unlikely to change over the subsequent iterations because all the available information has been yet used by the interpolation scheme. For the $i$th gridpoint, the critical scale can be defined as the spatial scale $\alpha$ where the last significant variation of the interpolated relative anomaly $\overset{\alpha}{\mathbf{x}}{}^a{}_i$ has occurred with respect to the value $\overset{\beta}{\mathbf{x}}{}^a{}_i$ (or equivalently $\overset{\alpha}{\mathbf{x}}{}^b{}_i$) obtained for the immediately preceding scale $\beta$. The critical scale is variable across the domain and depends on both the spatial structure of the precipitation field and the local station density. The smaller the critical scale, the

more noisy and rough is the analysis field. Typically, stratiform precipitation would lead to larger critical scales than convective precipitation. In any case, the small-scales determined locally by the observational network, as defined in the Introduction, pose lower limits to the critical scale.

## 4   Example application for precipitation

An example of application of the spatial interpolation method described in Sec. 3.2 is given in this section. We have chosen to illustrate the analysis procedure applied to the day 1998-10-24, that is one of the days within the period 1957-2017 with the highest value of averaged observed precipitation and where almost all the gauges have measured precipitation.

The reference field for October (see Sec. 2) is the prior information used in our spatial analysis and it is shown in Fig. 3. The highest precipitation values occur in the western part of Norway and a precipitation gradient between the coast and inland
areas is evident.

The first of the three fundamental steps in Eq. (10) is the iterative application of OI for the Box-Cox transformed relative anomalies over a sequence of decreasing spatial scales. We are looping over 91 scales and in Fig. 4 three of those scales are shown. As a result of the transformation, the analysis fields on the maps are not straightforward to interpret as they do not correspond to e.g. precipitation. For this reason, we have chosen an ad-hoc colour scale that highlights the patterns in the fields
more than the differences among values. For each scale, the OI is implemented as in Eqs. (11)- (13). The smaller the spatial scale, the more fine-scale details the OI will represent and, as a consequence, the finer is the grid used. For the scales of 100 km, 31 km and 2 km, the interpolation are performed over grids of 50 km, 15 km and 1 km, respectively. From the coarser scale represented in Fig. 4, it is only possible to have a rough idea of the final field as this is only the 17th iteration of the 91 totals. At the scale of 31 km, that is iteration 69, the main features of the field are easier to recognize: the largest values are in
the eastern part of the domain, where the reference values are smaller; in the middle of Southern Norway, the field reaches its minimum. The smallest scale is equal to 2 km. With reference to the station distribution in Fig. 1, the patterns become smaller in data-dense regions while the largest "blobs" occur in data-sparse regions. As one may expect, the patterns on the 1 km grid in Fig. 4 matches the spatial structure of the patterns in Fig. 2.

Figure 5 shows the critical scale defined in Sec. 3.2 for the day 1998-10-24. The critical scale is related to variations in the
analyses between subsequent scales of the multi-scale OI scheme presented in Eqs. (11)- (13). Examples of such analyses are the fields shown in Fig. 4. Specifically, for each gridpoint, the field in Fig. 5 shows the spatial scale where the last significant variation of the interpolated precipitation relative anomaly has occurred. In practice, with reference to Eqs. (11)- (13), the critical scale at the $i$th gridpoint is the largest scale $\alpha$ where the absolute value of the difference $\overset{\alpha}{\mathbf{x}}{}^a{}_i - \overset{\beta}{\mathbf{x}}{}^a{}_i$ is smaller than 0.01 (dimensionless units) or 1%. In particular, it is interesting to know where the precipitation field represents features at the
smallest spatial scales included in the multi-scale OI (i.e., from 2 km to 20 km). Those spatial scales correspond to the meso-$\gamma$ atmospheric scale and are suitable to proper represent e.g., thunderstorms (Thunis and Bornstein, 1996). It is less relevant to distinguish between spatial scales above 30 km and up to 200 km as they belong to the same meso-$\beta$ scale e.g, fronts, thunderstorm groups. Fig. 5 emphasizes the gradient between meso-$\gamma$ and meso-$\beta$ scales. In the data-sparse regions marked

as red in Fig. 2, the station network poses a constraint on the smallest scales the analysis can represent and, as a consequence, the critical scales belong to the meso-$\beta$ scale, regardless of the precipitation type. On the other hand, where the observational network is dense, the critical scale assume a broad range of values, that is from 2 km to 100 km, depending on the local characteristics of precipitation (e.g., stratiform or convective). In regions where the critical scale is within the meso-$\gamma$ scale, the precipitation field is highly variable over relatively short distances and this might indicate the occurrence of convective precipitation.

Fig. 6 shows the RR analysis field as derived from Eq.(10). It is a combination of the reference field shown in Fig. 3 and the multi-scale OI field shown in Fig. 4 for the scale of 2 km. In particular, the local variations of precipitation in almost data-void regions (e.g, some mountainous areas in Southern Norway that are marked as red regions in Fig. 2) are mostly due to the reference field.

## 5 Verification

The evaluation is based mostly on CV exercises and comparison against the seNorge2 datasets of RR and TG. The cross-validation analysis (i.e., CV-analysis) is the analysis value at a station location obtained considering a selection of the available observations that does not include the one measured at that location. If CV is applied systematically to all stations and it includes all the remaining observations then it is called leave-one-out cross-validation (LOOCV).

The summary statistics of the following variables are used: CV-analysis residuals (i.e, CV-analyses minus observations); innovations (i.e., background minus observations); and analysis residuals (i.e, analyses minus background). The CV-analysis, background and analysis are evaluated through the statistics of CV-analysis residuals, innovations and analysis residuals, respectively. Note that the background is not considered in the verification of precipitation. At a generic station location, CV-analysis and background are independent from the observation, while the analysis has been computed using the observation. As a consequence, the statistics of CV-analysis residuals and innovations have similar interpretations. The CV-analysis residual distributions are used in place of the unknown analysis error distributions at gridpoints. The innovation distributions are used to investigate the properties of the background error at gridpoints. On the other hand, the statistics of analysis residuals reveal the filtering properties of the statistical interpolation at station locations that are related to the observation representativeness error (Lussana et al., 2010; Lorenc, 1986; Desroziers et al., 2005). The mean absolute error (MAE) and the root mean square error (RMSE) quantify the average mean absolute deviation and the spread, respectively, of a variable.

For temperature, we are using LOOCV. The comparison between the statistics of CV-analysis residuals and innovations quantifies the improvement of the analysis over the pseudo-background at gridpoints. The fraction of errors (i.e., absolute deviations) greater than $3°C$ has been used as a measure of the tails of the distribution of deviation values. Note that the threshold of $3°C$ is used in MET Norway's verification practice to define a significant deviation that undermines the user's confidence in the forecast.

For precipitation, LOOCV is computationally too expensive. Thus, for each day a random sample of 10% of the available stations have been reserved for CV and they are not used in the interpolation. Because precipitation errors follow a multi-

plicative rather than an additive error model (Tian et al., 2013), large errors for precipitation are defined as absolute deviations between CV-analysis and observation greater than 50% of the observed value.

## 5.1 Temperature

### 5.1.1 Summary statistics of the verification scores at station locations

In Figures 7-8 the TG, TX and TN evaluation results are shown for summer and winter, respectively, as a function of CV-IDI (Sec. 2.3). The whole 61-year time period is considered. For each of the four CV-IDI classes (Sec. 2.3), the mean value of the score is displayed. The CV-analysis and background always score better in data-dense areas, as expected. The analysis evaluation scores do not vary much with respect to variations in station density, thus indicating that the observation representativeness error is rather constant for all stations. This result support the use of a single value characterizing the whole

observational network for $\varepsilon^2$ in Eq. (6).

For all variables, the spatial interpolation scheme generally performs better during summer than winter when small-scale processes (e.g., strong temperature inversion) are more frequent. The TG analysis error distribution at gridpoints, as estimated by CV-analysis residuals, shows that during the summer, the MAE is between $0.5°C$ and $1°C$ and its RMSE is also around $1°C$, and errors larger than $3°C$ are unlikely even in data-sparse regions; during winter, MAE and RMSE double their values

and the differences between data-dense and data-sparse regions are larger, with the probability of having large errors being approximately 25% in data-sparse regions. The TX analysis error behaves similarly to TG. Note that the spatial interpolation method during summer performs better on TG than on TX, while in winter the opposite occurs. The distribution of the TN analysis error at gridpoints has a much larger spread than those of TG and TX. The TN RMSE is between $1.5°C$ and $2°C$ in summer, and up to approximately $4°C$ during winter in data-sparse regions. The tail of the TN distribution is also longer and

large errors are more frequent than for the other daily temperatures. At the same time, the average bias (MAE) is also larger for TN.

### 5.1.2 seNorge_2018 and seNorge2 comparison of TG

The seNorge_2018 spatial interpolation procedure builds upon seNorge2. Several modifications have been made, though keeping the scale-separation approach. In seNorge_2018 a single function has been used to model the sub-regional vertical profile,

instead of the three different functions used in seNorge2. At the same time, in seNorge_2018 the blending of sub-regional fields into a regional pseudo-background field is based on a much larger number of sub-regional fields.

In Figure 9 the TG dataset is compared with seNorge2 over two multi-year time periods for the winter season. In the other seasons (not shown here) the differences are almost always between $-2°C$ and $1°C$. For most of the Norwegian mainland, seNorge_2018 is colder than seNorge2, with the larger differences on the mountain tops and along the coast in the North. The

two periods show similar patterns, however for the 1991-2015 period there are some regions where seNorge_2018 is warmer than seNorge2, such as the plateau in the North between Finland, Russia and Norway; along the border between Sweden and Norway; and in the mountains of Southern Norway.

The variations between seNorge_2018 and seNorge2 having the most significant impacts on the differences shown in Fig. 9 are in the OI settings. seNorge_2018 includes the land area fraction as an additional parameter in Eq. (7) and this improvement causes the differences along the coastline, where stations having significantly different land area fractions are less correlated between each other. By considering only stations along the coast and CV-analysis residual data from 1957 to 2017, seNorge2 has MAE=0.79°C and RMSE=1.12°C, while seNorge_2018 has MAE=0.7°C and RMSE=1.03°C. If compared to seNorge2, seNorge_2018 reduces the bias along the coast by 10% and the RMSE by 8%.

The evident differences between the two datasets are in the mountains, where seNorge_2018 often presents warmer valley floors and colder ridges. In particular, according to Fig. 9, the highest peaks of the Scandinavian mountains are on average up to 9°C colder, while the valley floors are a few degrees warmer. Those differences can be explained by (1) the reduced $D^z$ value in seNorge_2018 ($D^z = 210$ m instead of $D^z = 600$ m as in seNorge2), which narrows the vertical layer where OI adjustments are effective, therefore temperatures in data-sparse high-altitude regions mostly coincides with background values; (2) the modified procedure for the pseudo-background calculation that realizes a smoother transition between neighbouring sub-regional pseudo-background fields. By considering all the stations and CV-analysis residual data from 1957 to 2017, seNorge2 has MAE=0.81°C and RMSE=1.18°C, while seNorge_2018 has MAE=0.76°C and RMSE=1.18°C.

## 5.2 Precipitation

### 5.2.1 Summary statistics of the verification scores

In Figure 10, the RR dataset is evaluated through the statistics of CV-analysis residuals. In general, the spatial interpolation performs significantly better for station-dense areas, then the performances degrade faster for data-sparse regions as shown by the MAE and RMSE for observations greater than 1 mm/day. The ability in distinguishing between precipitation and no-precipitation is shown by the Equitable Threat Score (ETS) with a threshold of 1 mm/day. In station-dense areas the fraction of correct predictions, accounting for hits due to chance, is approximately 70%, while the ETS goes down to 0.4 in data-sparse regions. With respect to intense precipitation (i.e., observed values greater than 10 mm/day), the probability of having large errors is less than 5% in data-dense areas and around 30% in data-sparse areas.

The close relationship between the terrain and the annual total precipitation is shown in Fig. 11, where values up to 4700 mm/year are reconstructed along the Norwegian coast.

### 5.2.2 seNorge_2018 and seNorge2 comparison

Figure 11 shows the differences between seNorge_2018 and seNorge2 mean annual total precipitation in the period 1957-2015. On average, seNorge_2018 has significantly higher precipitation values than seNorge2, especially in data-sparse mountainous regions, and this is due to: i) the correction of rain-gauge data for the wind-induced under-catch, that increase the observed precipitation mostly during winter, and ii) the modified statistical interpolation scheme that uses more information in data-sparse regions, which often tends to increase the precipitation analysis when compared to seNorge2.

For precipitation, we have stated in the Introduction that seNorge_2018 uses a reference to increase the effective resolution of the field, compared to the resolution given by the spatial distribution of the observational network alone. In the context of forecast verification, Casati et al. (2004) decompose precipitation fields on different spatial scales using a two-dimensional discrete Haar wavelet decomposition. As described in the paper by Casati (2010), this wavelet decomposition can be used to

5 study the average energy per cell (hereafter energy) of the scale components of precipitation, where the energy is defined as "the average of the field gridpoint squared value". Fig. 12 shows the mean energy of the scale components for seNorge_2018 and seNorge2 based on the 25% of cases with the higher values of averaged precipitation over the domain within the time period 1957-2017. The left panel shows the energies, where seNorge_2018 has more energy than seNorge2 for all scales, a result that is in agreement with Fig. 11. The shapes of the two energy distributions are similar, with a maximum at the scale of 128 km

and a gradual decrease of the energy for smaller spatial scales. However, the right panel shows that the energy percentages for seNorge_2018 are systematically larger than for seNorge2 for scales smaller than 128 km. Because the effective resolution of seNorge2 is determined by the observational network only, the results presented in Fig. 12 support our initial statement on the increased effective resolution of seNorge_2018.

## 5.3 Occurrence of large errors as a function of the station density

Next we discuss more in detail the relationships between the occurrence of large errors, as they have been defined at the beginning of Sec. 5, and the observation densities. We have investigated those relationships for both temperature and precipitation aiming at quantifying the impacts on the analysis quality due to variations in the station distribution.

In Figure 13, the expected percentage of large errors at gridpoints is shown for all variables. In the case of temperature, the season of the year is an important factor in determining the quality of the analyses and in Fig. 13 we have considered only the

20 winter seasons, that is when errors are larger. For precipitation, the season of the year is a less compelling factor, compared to the station density, in determining the quality of the analysis. Then, the whole RR dataset is taken into account in Fig. 13. The close relationships between CV-IDI and the occurrence of large errors are evident from the scatter-plots in Fig. 8 (bottom row) and Fig. 10 (panel d). The same relationships are also represented in the scatter-plots in the panels on the bottom left of each of the maps in Fig. 13. Note that for summertime temperatures, the relationship between CV-IDI and occurrence of large errors

is shown in Fig. 7 (bottom row). If we define the occurrence of large error (err) as a function of CV-IDI ($x$) as $\mathrm{err}(x)$, then the points in the scatter-plots of Fig. 13 represent the actual occurrence of large errors at station locations and the lines are the best fit to the points of the Gaussian function:

$$\mathrm{err}(x) = a \, \exp\left[-0.5\frac{(x-b)^2}{c^2}\right] \tag{15}$$

The estimated large error occurrence at gridpoints is obtained by replacing the CV-IDI with the IDI fields (Fig. 2) in Eq. (15).

The three parameters of the bell curve shape are: the value of the curve's peak $a$; the position of the center of the peak $b$, and $c$ which controls the width of the bell. The Gaussian function provides reasonably good fits to the points, the relationship between the station density and the analysis quality is non-linear and the analysis performances decay much faster in data-sparse than in data-dense regions. The parameters values for the three variables are: TG $a = 703.012$, $b = -2.626$, $c = 1.123$;

TX $a = 856.420$, $b = -2.924$, $c = 1.114$; TN $a = 448.39$, $b = -2.891$, $c = 1.431$. The probability of having a large analysis error is remarkably small over the domain for TX, while for TN such large errors are quite common. The situation for TG is somewhat in between those two extreme cases and large regions of the domain are unlikely to present large errors. Once again, it is evident that the worst performances occur in those regions characterized by complex terrain and low station density, such

as the mountainous region between Norway and Sweden in the northern part of the domain.

As for temperature, the expected percentage of large errors over the precipitation grid is shown in Fig. 13. The three parameters of Eq. (15) for RR are: $a = 50.438$, $b = -0.676$, $c = 0.846$. The elevation is not considered in the IDI definition for RR, so the map looks rather different than the temperature maps. Furthermore, by setting $D = 10$ km in Eq. (4) (instead of $D = 50$ km as for temperature) we have imposed a fast transition between data-dense and data-sparse regions. For data-dense regions, the

expected percentage of large analysis errors for intense precipitation is less than 10%. For data-sparse regions, the percentage of large errors is approximately 40%.

## 6  Discussion

Because the presented statistical interpolation methods automatically adapt to the local observation density, the user of the seNorge_2018 dataset must be aware that: (i) the comparison between different sub-regions over the domain is influenced

by the respective local station densities, and (ii) variations in the observational network over time will affect temporal trends derived from this dataset (Masson and Frei, 2016).

For the four variables, we have investigated the variations of the performances of our interpolation schemes between two different time periods, 1961-1990 (61-90) and 1991-2015 (91-15). The evaluation scores are similar to the ones presented in Sec. 5 for the whole 61-year time period, thus indicating that the ongoing climate change does not have a significant impact on

the reconstruction skills of our statistical interpolation schemes.

The three main factors determining the quality of the temperature datasets are the season of the year, the station density and the terrain complexity. The last two factors are correlated, as shown by Figs. 1- 2. TN is the most challenging variable to represent, possibly because atmospheric processes at unresolved spatial scales occur more frequently for this variable than for the others. It is worth pointing out that Fig. 7 shows that during the summer TN has a smaller representativeness error in

data-sparse than in data-dense regions, even though the differences are rather small. It is not clear whether this is due to the spatial interpolation scheme or to the occurrence of specific atmospheric phenomena (e.g, urban heat island effect). In general, our results indicate that a denser station network is needed to deliver TN products having the same quality as TG and TX by using the statistical interpolation method presented in Sec. 3.1. Future developments of the method may also improve the representation of TN, such as: (1) consider the differences between TG and TN, instead of TN; (2) develop a relationship for

the vertical profile of TN, that would replace the one proposed in Sec. 3.1; (3) use numerical model output fields (e.g., from reanalysis) in addition to the observed data.

The two main factors determining the quality of the precipitation dataset are the station density and the terrain complexity. The season of the year seems to have a smaller impact on the verification scores.

With respect to seNorge2, seNorge_2018 presents several methodological improvements and two additional variables, TX and TN. Furthermore, it should be mentioned that there had been variations in the observational datasets used for the production of the two gridded datasets. Even though the data sources are the same for both seNorge datasets, seNorge2 is based on an observational dataset that has been produced in 2016, while seNorge_2018 benefits from the latest efforts in data collection and quality control made by MET and the ECA&D team. The verification results show that seNorge_2018 temperature predictions are on average more accurate than seNorge2, especially along the coast. With respect to the predictions of precipitation, as described in the seNorge2 paper Lussana et al. (2018a), this dataset is likely to underestimate precipitation so we have designed seNorge_2018 to returns higher precipitation values because this would better agree with the Norwegian water balance and eventually improve the results of hydrological simulations based on seNorge_2018 than for seNorge2. This last point needs to be verified in the near future. In Sec. 5.2.2, it has been demonstrated that seNorge_2018 has an higher effective resolution than seNorge2. In conclusion, we recommend the use of seNorge_2018 instead of seNorge2 for both TG and RR.

## 6.1   TG, TX and TN cross-checking

The cross-checking is mentioned at the beginning of Sec. 3.1 and it is done on the analyses. TG is used as a reference value for both TX and TN, since: (a) the construction of the pseudo-background field is based on a vertical profile relationship that has been developed for TG; (b) TG is expected to be a more robust variable than TX and TN (in the sense defined by e.g., (Lanzante, 1996)), because it is an averaged value instead of an extreme. For each gridpoint, we check whether TN (TX) is smaller (greater) than TG. Wherever the condition is not satisfied, TN (TX) is replaced by TG.

Because of the 12-hour offset in the definitions of TG and either TX or TN, the cross-checking can be wrong. Nevertheless, it is a useful check to identify those situations where the interpolation of daily extremes is not convincing. In fact, despite the offset in the definitions of the 24-hour aggregation period, for a typical day TN is smaller than TG and TX is greater than TG. We have found very rare exceptions to these rules in the surroundings of station locations. In the vast majority of cases, the cross-checking flags those gridpoints in the mountains (i.e., far from station locations) where the extrapolation of the vertical profile cannot be adjusted by means of observed data, which are located in large part on the valley floors (Fig. 1). As a result, the simulated daily extremes are affected by significant uncertainties and we use TG as a reference value to assess the significance of those uncertainties. When the cross-check reports possible violation of the physical consistency, our choice is to "mask out" the analyses of daily extremes and at the same time we propose TG as an alternative.

The number of gridpoints flagged by the cross-checking vary seasonally and it is higher in winter and lower in summer. In the case of TN, the cross-checking flags on average 9% of the gridpoints in winter and 1% in summer. In the case of TX, the cross-checking flags on average 7% of the gridpoints in winter and 1% in summer.

Future developments will focus on improving the cross-checking, in order to properly handle those exceptional situations that are currently erroneously flagged as physical inconsistencies among the three variables.

## 7 Conclusions

seNorge_2018 provides 61-year (1957-2017) datasets of daily mean, maximum, and minimum temperatures, as well as daily total precipitation, over Norway and parts of Finland, Sweden and Russia. The plan at MET Norway is to update the historical dataset once a year, while at the same time provisional daily estimates for the current year are computed every day. MET Norway has an open data policy and all the datasets, as well as most of the observations used in the calculations, are available for public download via its web services.

The observational datasets have been obtained through statistical methods that build upon our previous works. The interpolation schemes automatically adapt their settings to the local station density and this allows for a higher effective resolution in data-dense areas, while in data-sparse regions the analysis is always the estimate of at least a few stations.

The main factor determining the quality of the temperature analysis are: the season of the year, the station density and the terrain complexity. In the case of precipitation, those factors are: the station density and the terrain complexity. Because of the importance of the combination of station density and terrain, we have widely used the IDI concept in our evaluation.

The new seNorge_2018 shows significant differences when compared to its predecessor seNorge2, both for TG and especially for RR. While first qualitative evaluations indicate that this is an improvement, an indirect evaluation where seNorge_2018 would be used as the forcing data for snow- and hydrological modeling is needed to confirm this.

seNorge_2018 is MET Norway's first observational dataset providing TX and TN from 1957. The temperature analysis has the largest errors during winter and the TN is the most challenging variable to represent. For TG and TX, large analysis errors are expected only in winter and limited to almost data-void areas such as mountain tops. TN may present large analysis errors more often than TG and TX and for larger portions of the domain, especially in mountainous regions.

To fill commonly occurring spatial gaps for RR in data-sparse regions, the interpolation uses monthly fields of a high-resolution numerical model and adjusts this to an optimal fit with the measurements that are available in the area. As a result, seNorge_2018 has a finer effective resolution than seNorge2. The ability of the method to correctly distinguish between precipitation and no-precipitation depends critically on the station density. In the North, the sparser observational network is associated with a high occurrence of large analysis errors. The evaluation shows that large analysis errors are unlikely in the data-dense regions of Southern Norway, even for intense precipitation.

## 8 Code and data availability

The spatial interpolation software is available at (DOI:https://doi.org/10.5281/zenodo.2022479, Lussana, 2018a).

The open-access datasets are available for public download at:

– daily total precipitation (DOI:https://doi.org/10.5281/zenodo.2082320, Lussana, 2018b)

– daily mean temperature (DOI:https://doi.org/10.5281/zenodo.2023997, Lussana, 2018c)

– daily maximum temperature (DOI:https://doi.org/10.5281/zenodo.2559372, Lussana, 2018e)

– daily minimum temperature (DOI:https://doi.org/10.5281/zenodo.2559354, Lussana, 2018d)

seNorge_2018 is daily updated by MET Norway and the most recent data are available at http://thredds.met.no/thredds/catalog/senorge/seNorge_2018/catalog.html. Furthermore, it is possible to access the maps of precipitation or temperature analyses via:

– xgeo.no. Select *All Data* from the main menu, then *Weather*, then *seNorge_2018 precipitation* or *seNorge_2018 temperature*.

– http://thredds.met.no/thredds/catalog/senorge/seNorge_2018/catalog.html. Select a file (e.g., Latest/seNorge2018_20190815.nc), then *Viewers* and *Godiva2*.

*Competing interests.* The authors declare that they have no conflict of interest.

*Acknowledgements.* This research has been partially funded by the Norwegian project "Felles aktiviteter NVE-MET tilknyttet nasjonal flom- og skredvarslingstjeneste".

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

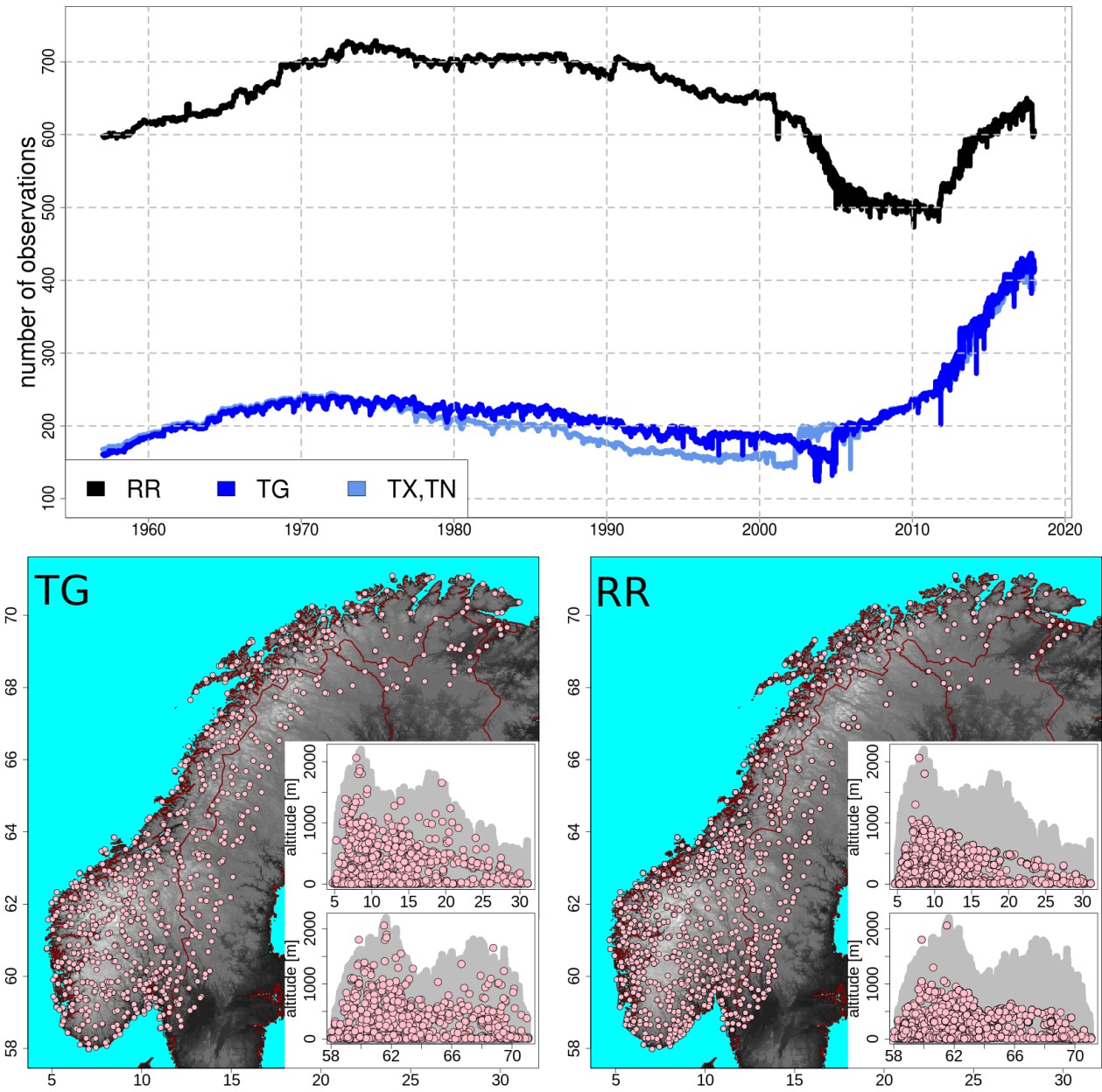

**Figure 1.** The observational network for the four variables: RR, TG, TX and TN. The top panel shows the time series for the number of available observations over the Norwegian mainland. The bottom panels show the observational networks for TG (left) and RR (right). TX and TN are not shown because they are similar to TG. The pink dots mark station locations with more than 1 year of data. In the bottom panels, the geographic coordinate system is used and the maps show the domain with topographic information derived from a high-resolution digital elevation model (DEM). For each of the two bottom panels, the two inset graphs show altitude above mean sea level as a function of latitude (top graph) and longitude (bottom graph). In each inset graph, the gray area shows the altitude of the terrain at gridpoints, while the pink dots are the altitudes of stations.

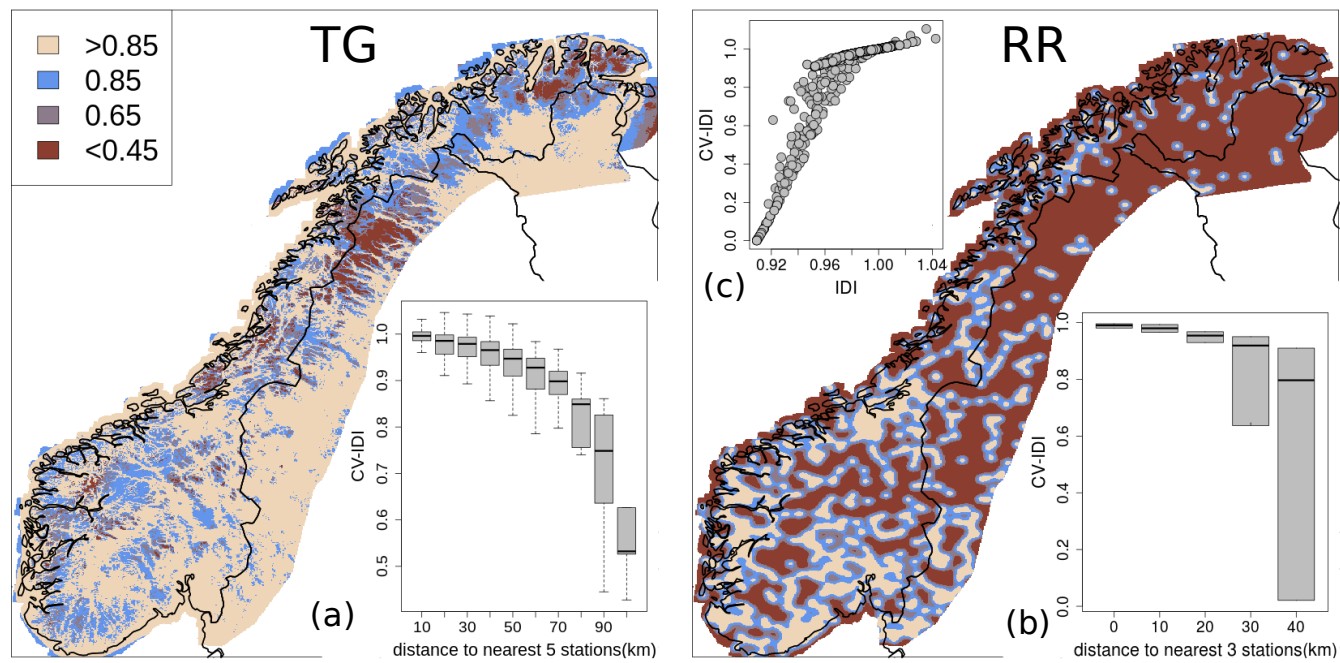

**Figure 2.** A representation of the observational network useful for spatial analysis of: TG (left) and RR (right). As for Fig. 1, TX and TN are not shown because they are similar to TG. The displayed fields are the Integral Data Influence (IDI) at gridpoints (same color scale for both fields). Panels *a* (TG, inset) and *b* (RR, inset bottom) show the Cross-Validation IDI (CV-IDI) as a function of the distance to the nearest stations. The distribution of CV-IDI is shown by boxplots, where each gray box represents a sample distribution: the black thick horizontal line is the median; the gray box width is the interquartile range; the whiskers extend to the tails. In panel *c* (RR, inset top), CV-IDI and IDI are compared and CV-IDI is shown as a function of IDI.

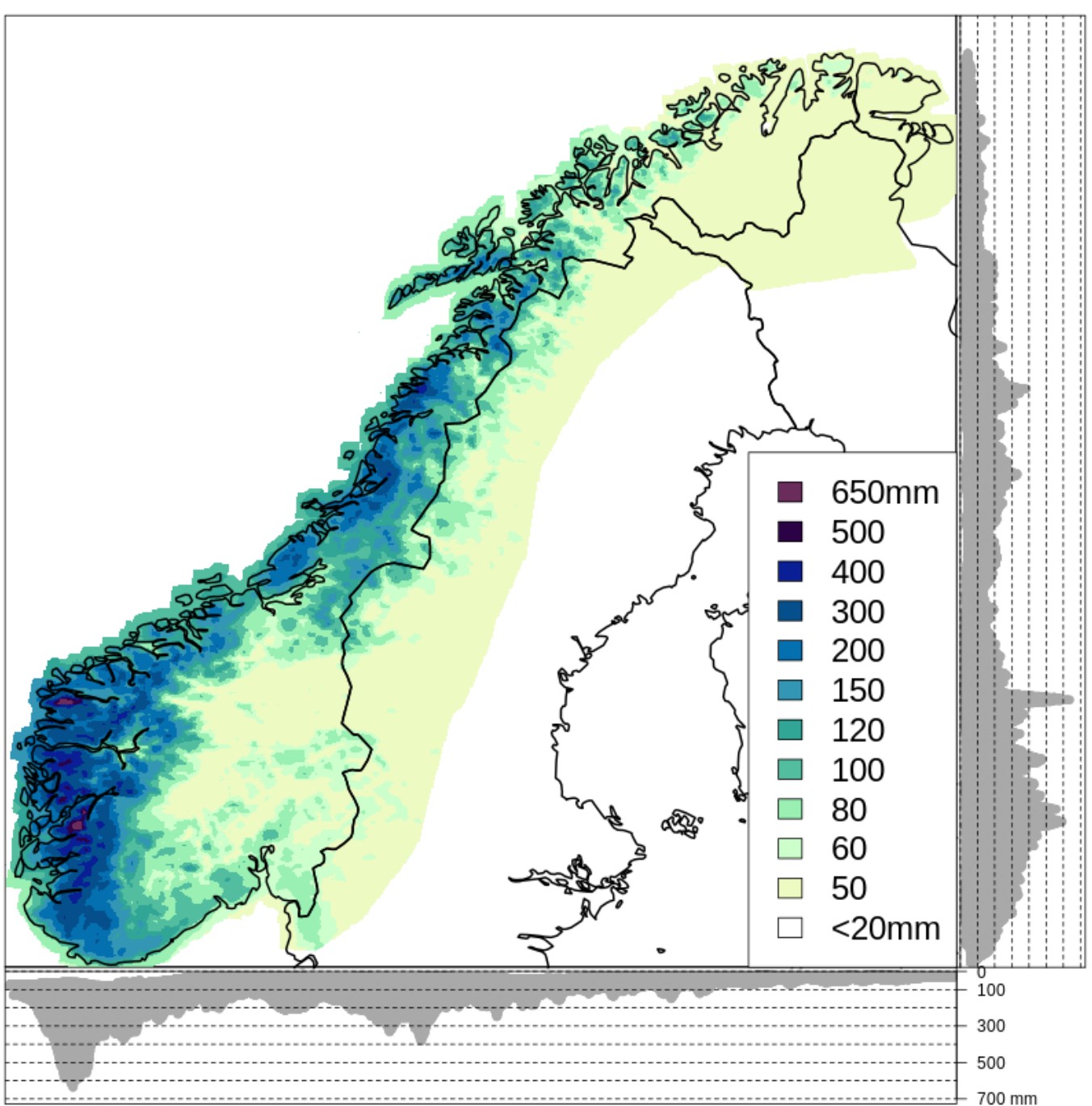

**Figure 3.** Precipitation reference field for October, that is $\mathbf{x}^{\text{ref}}$ in Eq. (10), used for the spatial analysis of RR for the day 1998-10-24 (Sec. 4). The lateral and bottom panels in both graphs show the projection of the reference precipitation values at gridpoints on the y- and the x- axis, respectively.

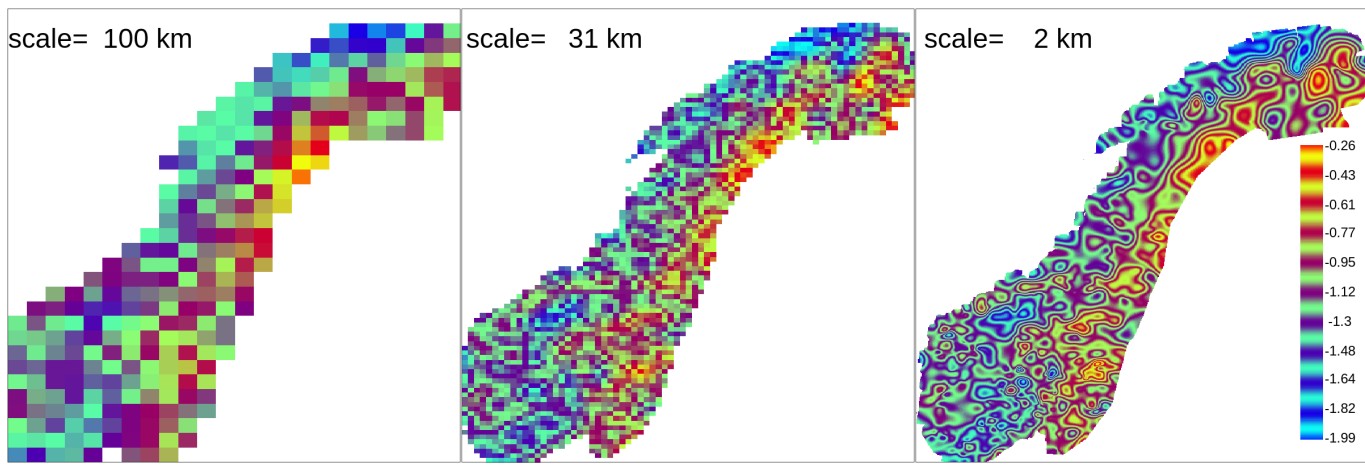

**Figure 4.** 1998-10-24 RR. Statistical interpolation of relative anomalies (dimensionless units) over different spatial length scales with the scheme reported in Eqs. (11)- (13). The interpolation loops over a sequence of 91 decreasing spatial scales: 100 km is the seventeenth; 31 km is the sixty-ninth; 2 km is the ninety-first. The fields shown are three different $\mathbf{x}^a_{\alpha}$ (Eq. (13)) for: $\alpha = 100$ km, $\alpha = 31$ km and $\alpha = 2$ km. In each of the the three cases, the interpolation is performed over a regular grid with the resolution of the integer value nearest to $\alpha/2$, that is: 50 km, 15 km and 1 km. The colour scale highlights the details in the field and it is the same for the three maps.

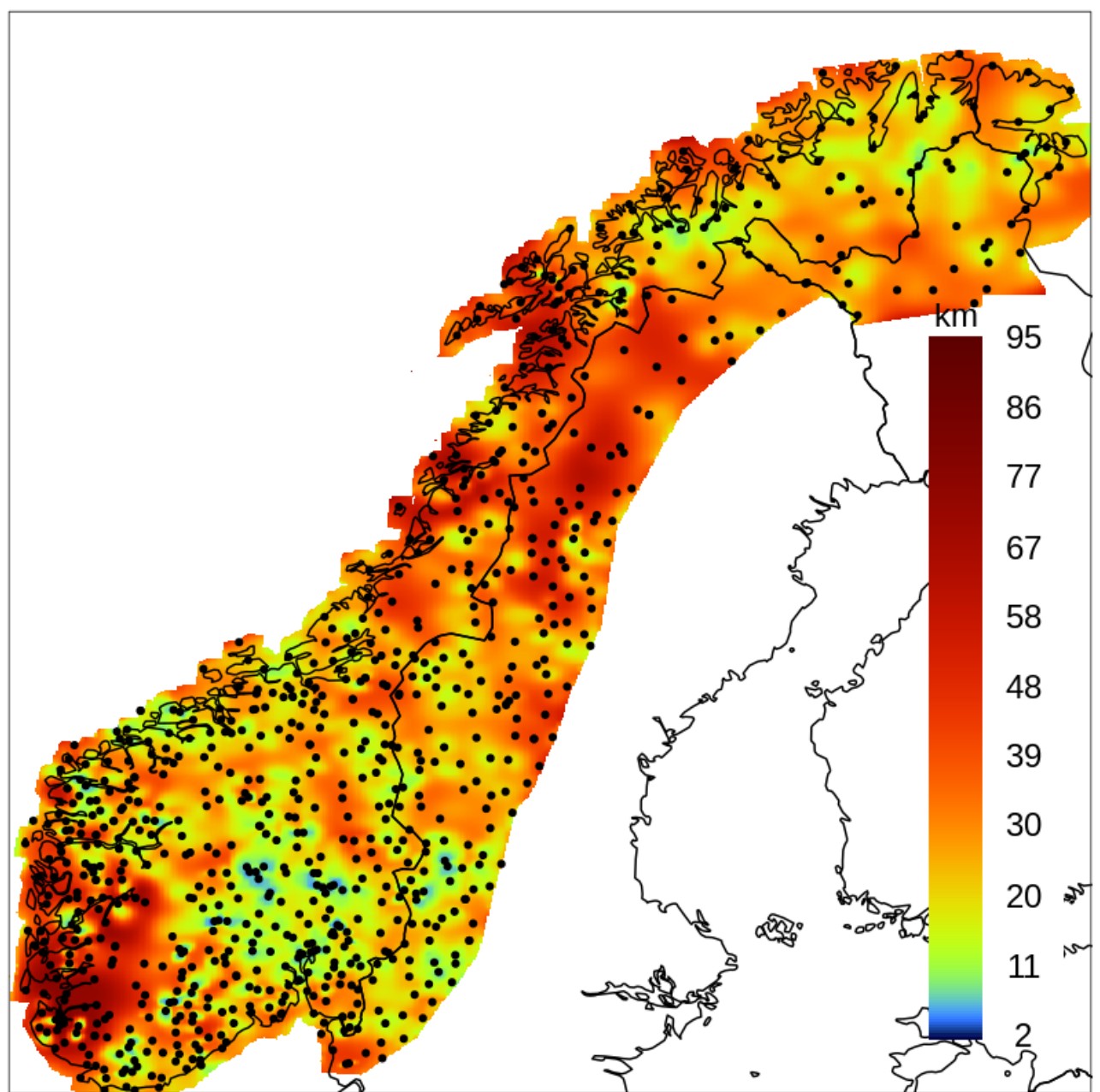

**Figure 5.** 1998-10-24 RR. "Critical" scale (Sec. 3.2; shaded field, values are in km) for the statistical interpolation scheme reported in Eqs. (11)- (13) given the observational network used (black dots mark the station locations). The critical scale varies for each gridpoint. For the $i$th gridpoint, the critical scale is the spatial scale $\alpha$ where the last significant variation of the interpolated relative anomaly $\overset{\alpha}{\mathbf{x}}{}^a_i$ has occurred with respect to the value $\overset{\beta}{\mathbf{x}}{}^a_i$ (or equivalently $\overset{\alpha}{\mathbf{x}}{}^b_i$) obtained for the immediately preceding scale $\beta$. A variation is considered significant when it is larger than 0.01 (dimensionless units) or 1%. The largest values of the critical scale may indicate either a data-sparse region or a region where stratiform precipitation is occurring. On the other hand, the smallest values occur in data-dense regions and they indicate regions where the field is highly variable over short distances (e.g., thunderstorms).

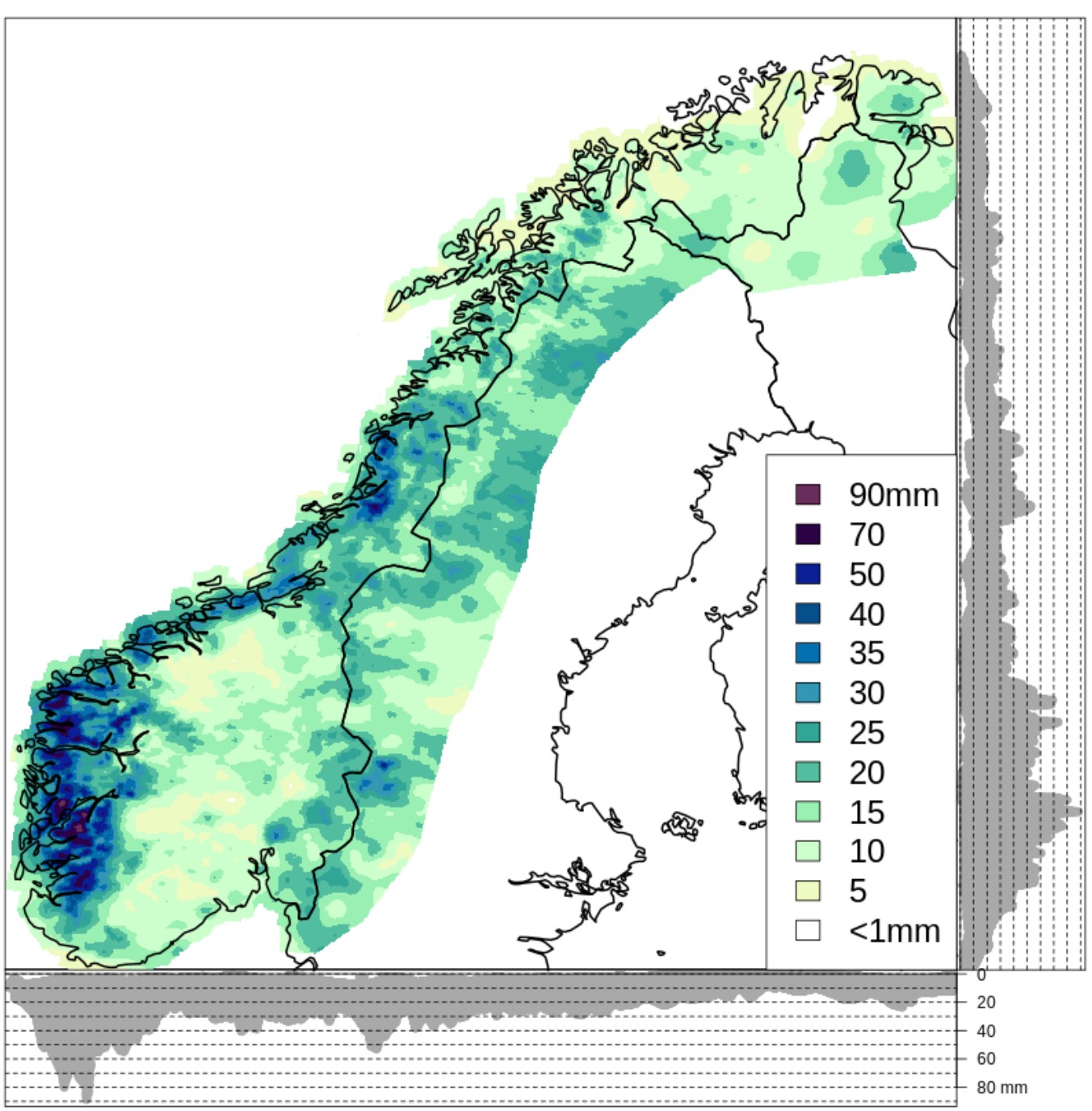

**Figure 6.** 1998-10-24, RR analysis field $\mathbf{x}^a$ of Eq. (10). The lateral and bottom panels in both graphs show the projection of the differences on the y- and the x- axis, respectively.

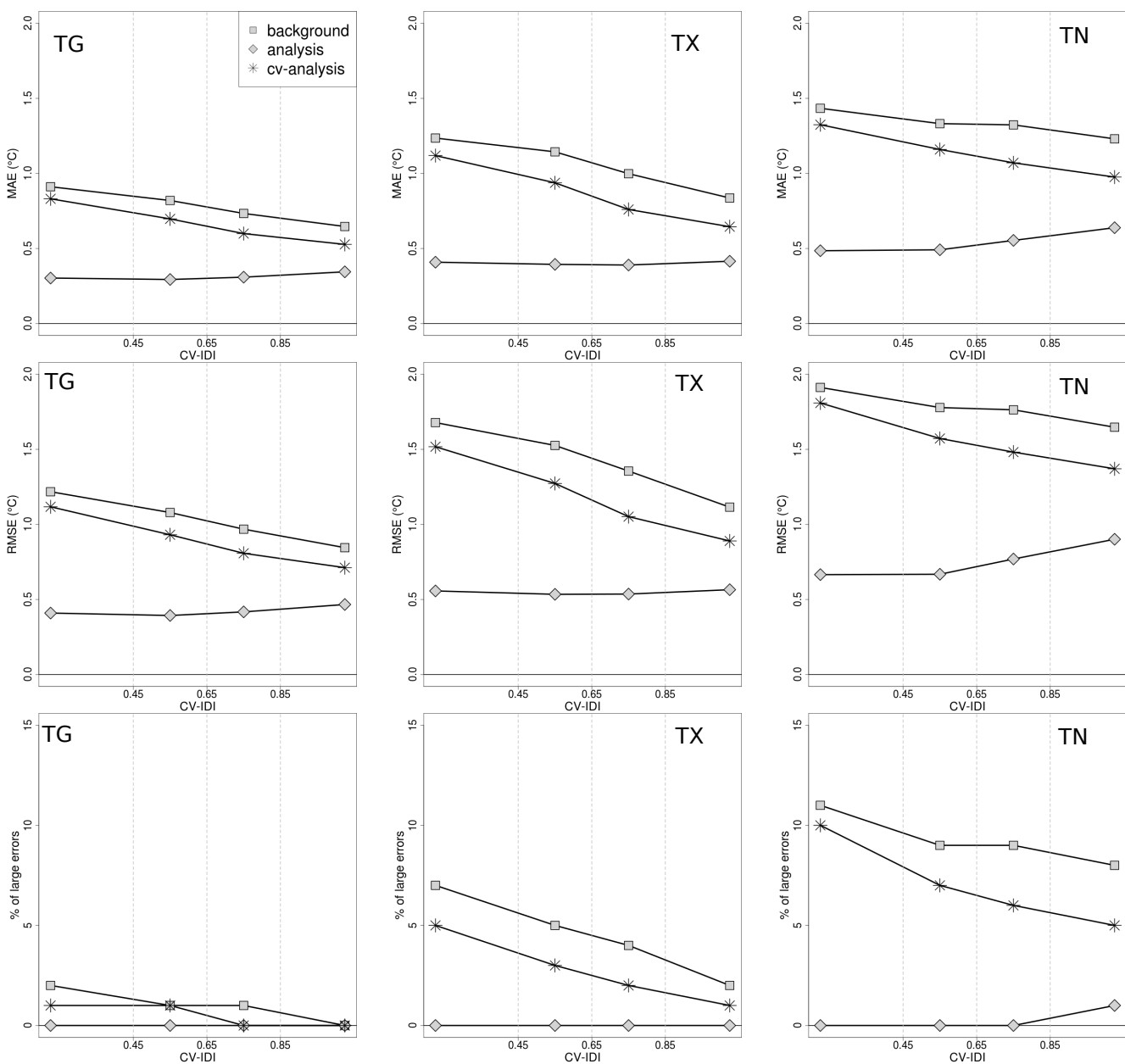

**Figure 7.** TG, TX and TN, verification scores as a function of CV-IDI for the summer seasons (June-July-August) of the 61-year time period 1957-2017. With reference to the definitions introduced in Sec. 5, the scores: for the analysis are based on the analysis residuals; for the CV-analysis on the CV-analysis residuals; for the background on the innovation. On the top row, the mean absolute error (MAE). In the middle, the root mean square error (RMSE). On the bottom row, the percentage of large errors. A large error is defined as the absolute value of innovation or residual larger than $3°$C.

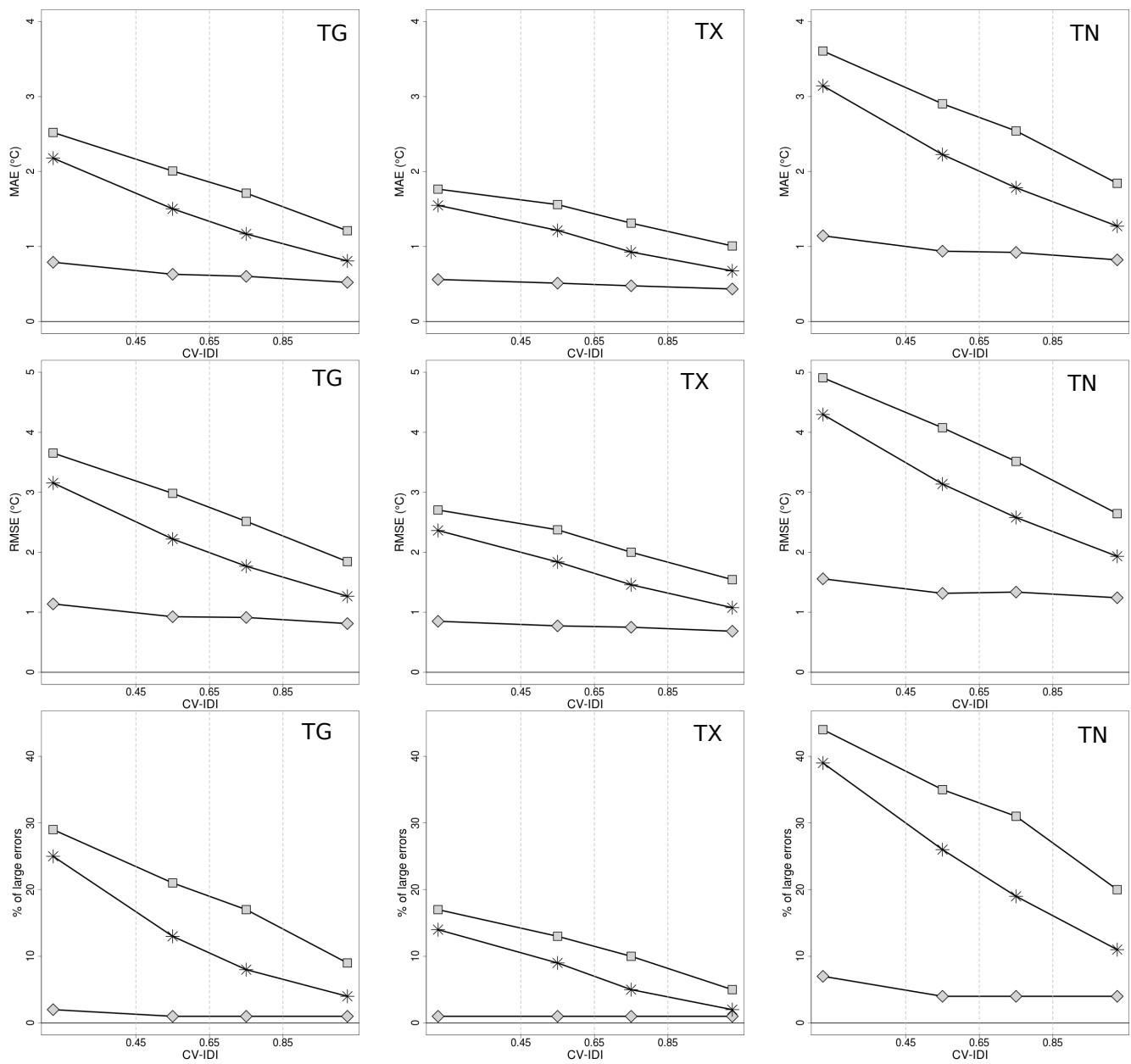

**Figure 8.** TG, TX and TN, verification scores as a function of CV-IDI for the winter seasons (December-January-February) of the 61-year time period 1957-2017. See Fig. 7 caption for further details.

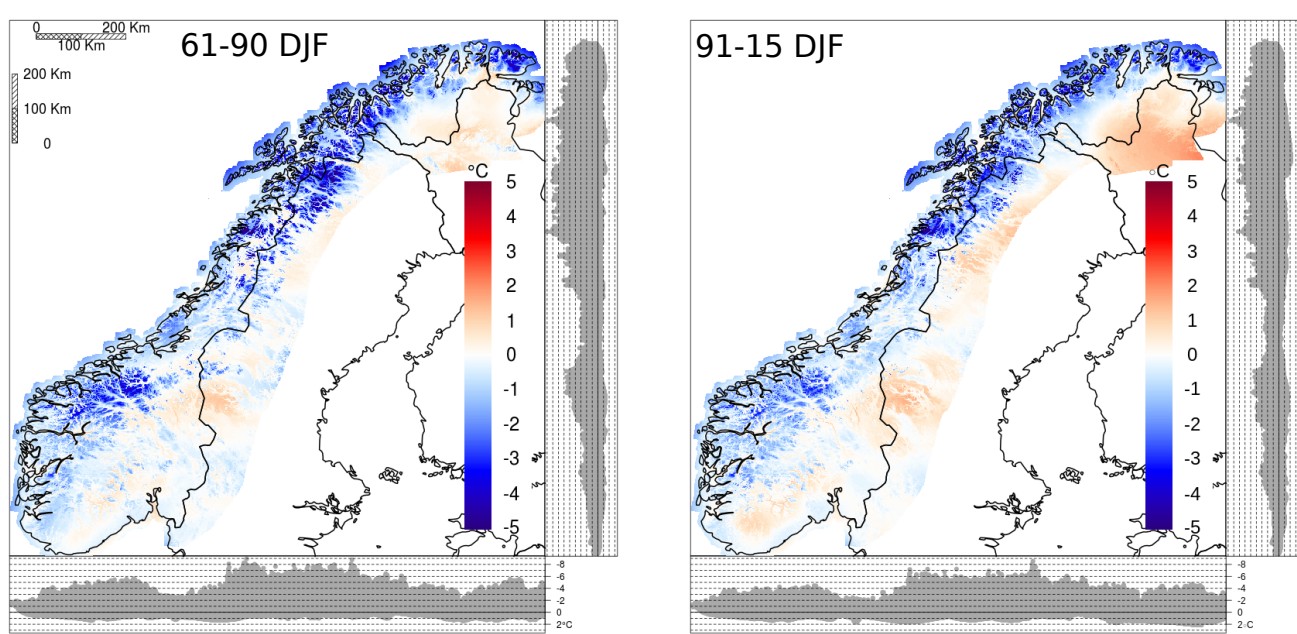

**Figure 9.** TG, mean difference between seNorge_2018 and seNorge2 daily analysis during winter (December-January-February). On the left panel, the 30-year period 1961-1990 is considered. On the right, the mean is based on the 25-year period 1991-2015. The lateral and bottom panels in both graphs show the projection of the values at gridpoints on the y- and the x- axis, respectively.

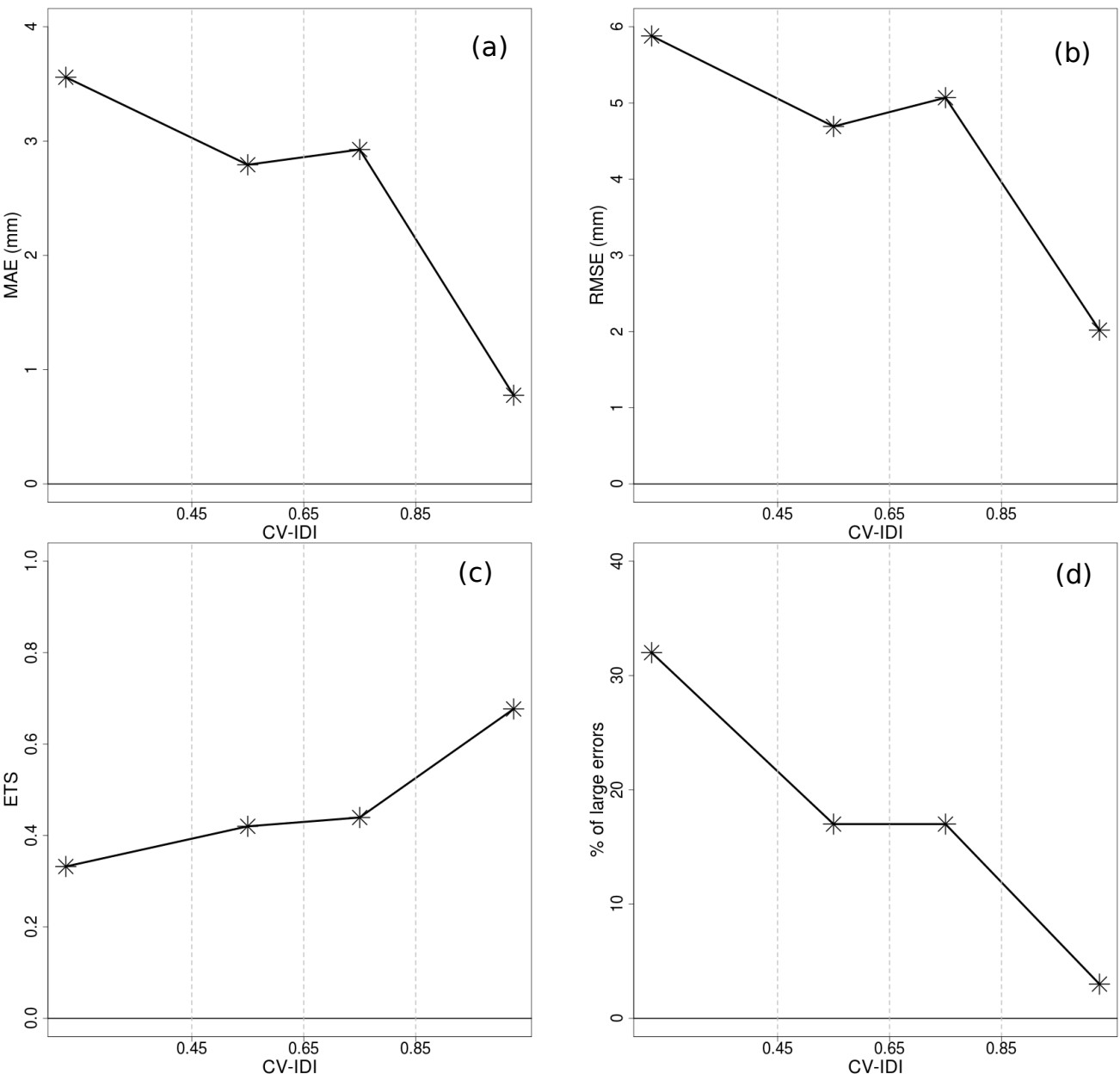

**Figure 10.** RR, CV-analysis verification scores as a function of CV-IDI for the 61-year time period 1957-2017. The terminology used is introduced in Sec. 5. Panel a, Mean Absolute Error (MAE) considering only observations greater than 1 mm/day. Panel b, Root-Mean-Squared Error (RMSE) considering only observations greater than 1 mm/day. Panel c, Equitable Threat Score (ETS) with threshold equals to 1 mm/day. Panel d, Percentage of large errors in case of intense precipitation. Intense precipitation is defined as an observed value greater than 10 mm/day. A large error is defined as the absolute value of a CV-analysis residual larger than 50% of the observed value.

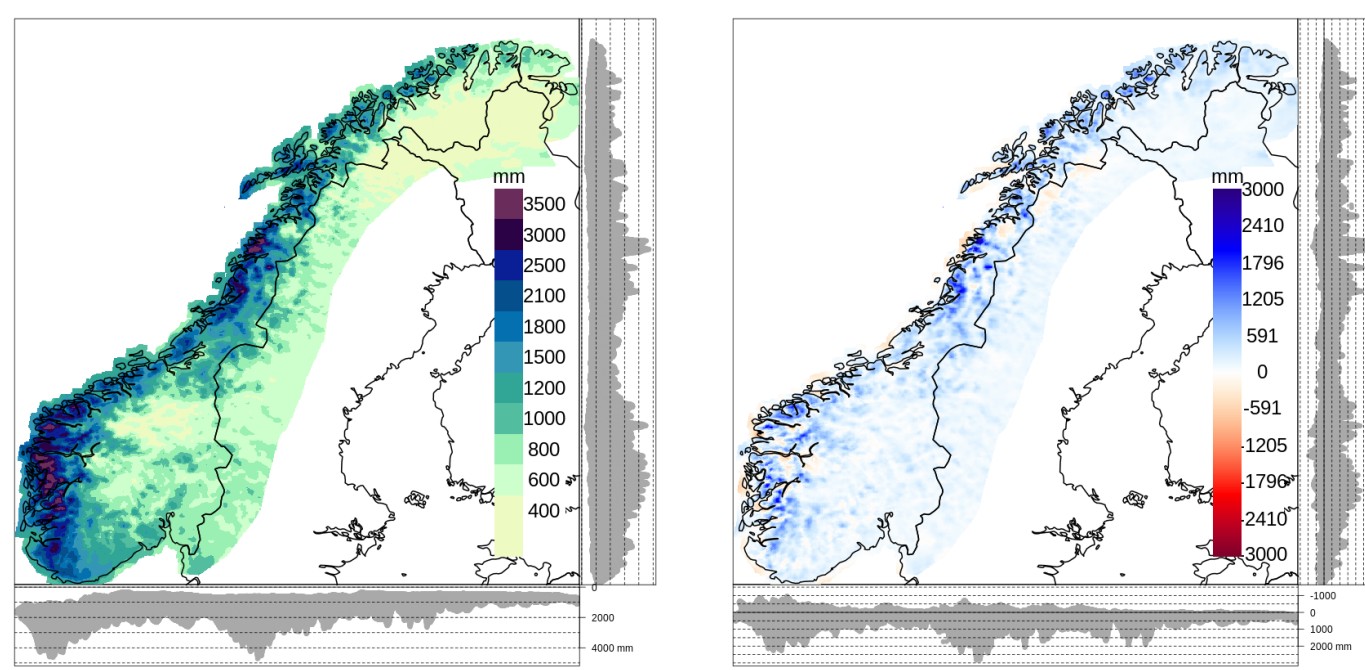

**Figure 11.** Annual total precipitation as derived by summing RR. On the left, the mean annual total precipitation based on the 61-year period 1957-2017: the lower precipitation class includes values smaller than 400 mm; the upper precipitation class values between 3500 mm and 4700 mm. On the right, mean annual total precipitation difference between seNorge_2018 and seNorge2 based on the 51-year period 1957-2015. On each graph, the lateral and bottom panels show the projection of the values at gridpoints on the y- and the x- axis, respectively.

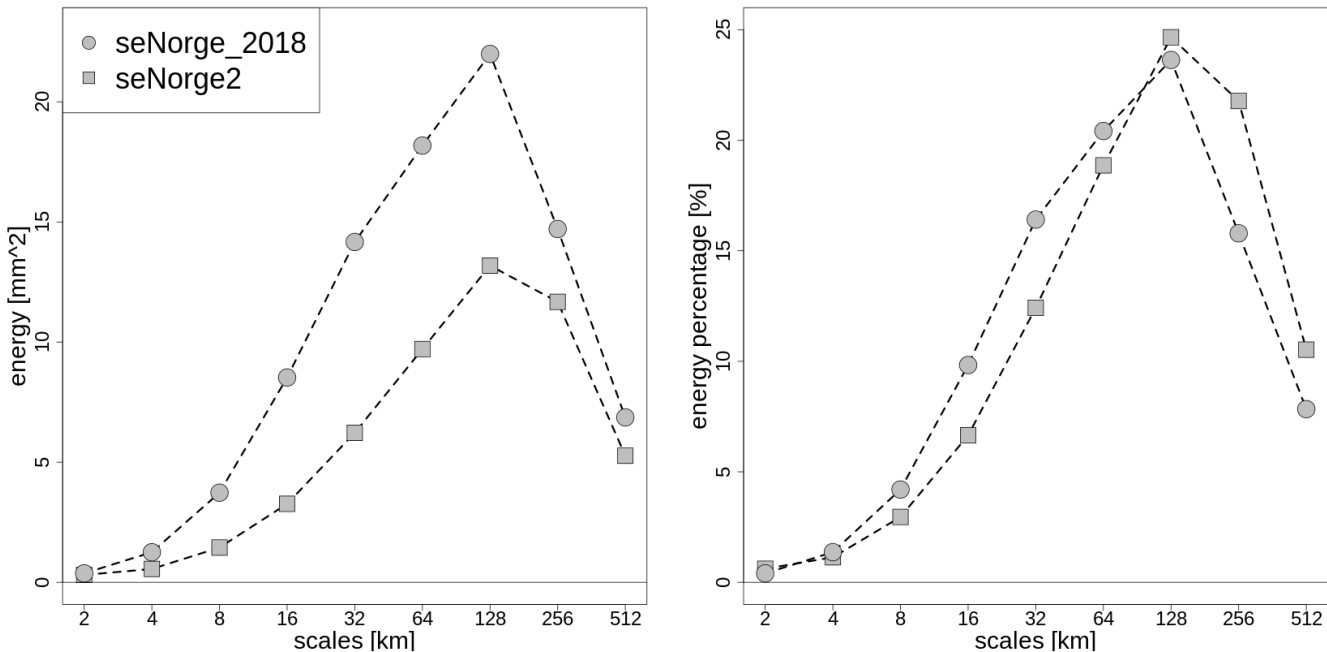

**Figure 12.** Scale decomposition of precipitation energy based on 2D discrete Haar wavelet transformation (Sec. 5.2.2). On the left, the averaged energy as a function of the spatial scale for seNorge_2018 and seNorge2. On the right, the averaged percentage of energy as a function of the spatial scale for the same datasets. The statistics is based on the 25% of cases with the highest values of averaged precipitation over the domain.

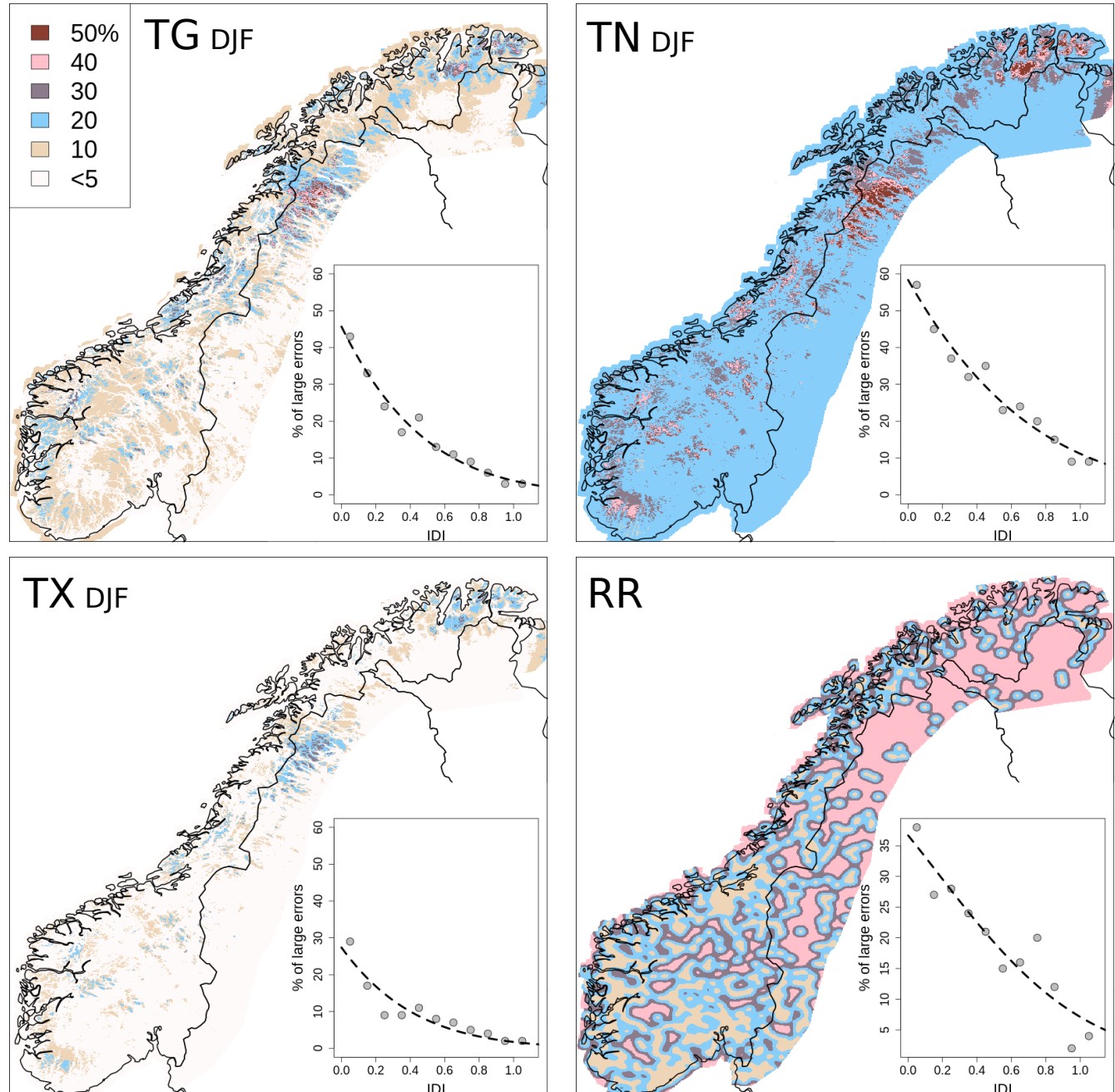

**Figure 13.** Expected percentages of large errors on the grid (dimensionless units) based on the summary statistics of the analyses and the spatial distribution of stations of the observational network. The colour scale is the same for all the maps. Wintertime (DJF) temperatures are considered, large errors are defined as deviations between analysis and unknown truth larger than $3°C$. All precipitation data has been considered, large errors are deviations between analysis and unknown truth larger than 50% of the analysis value when the analysis value is greater than 10 mm/day. The insets show the relation between IDI and percentage of large errors: the dots correspond to the percentage of large errors observed at station locations (on the x-axis, CV-IDI instead of IDI); the dashed lines are the best-fit functions used to infer the expected percentage of large errors at gridpoints.

**Table 1.** TG annual statistics: "n" is the average number of stations; "d" (km) is the average distance between a station and its nearest third station; $D^h$ (km), $D^z$ (m), $\sigma_b^2$ $((^\circ C)^2)$ and $\sigma_o^2$ $((^\circ C)^2)$ are the spatial interpolation parameters defined in Sec. 3.1. For each year, the optimal values of the interpolation parameters are obtained by imposing the constraint $\sigma_o^2/\sigma_b^2 = 0.5$ and considering the 1-year statistics of the innovation (observation minus background, Sec. 5).

| year | n | d | $D^h$ | $D^z$ | $\sigma_b^2$ | $\sigma_o^2$ |
|------|-----|----|----|-----|------|------|
| 1960 | 398 | 55 | 60 | 206 | 2.24 | 1.12 |
| 1970 | 669 | 42 | 45 | 217 | 1.86 | 0.93 |
| 1980 | 639 | 42 | 58 | 201 | 2.45 | 1.22 |
| 1990 | 600 | 44 | 57 | 202 | 1.33 | 0.66 |
| 2000 | 627 | 44 | 55 | 206 | 1.28 | 0.64 |
| 2010 | 639 | 45 | 52 | 206 | 2.45 | 1.23 |

**Table 2.** TX annual statistics. See Tab. 1 caption for further details.

| year | n | d | $D^h$ | $D^z$ | $\sigma_b^2$ | $\sigma_o^2$ |
|------|-----|----|----|-----|------|------|
| 1960 | 395 | 55 | 56 | 207 | 2.09 | 1.05 |
| 1970 | 669 | 42 | 57 | 201 | 1.67 | 0.84 |
| 1980 | 616 | 45 | 37 | 216 | 2.09 | 1.05 |
| 1990 | 563 | 47 | 55 | 206 | 1.40 | 0.70 |
| 2000 | 596 | 46 | 56 | 210 | 1.32 | 0.66 |
| 2010 | 638 | 45 | 57 | 206 | 2.09 | 1.04 |

**Table 3.** TN annual statistics. See Tab. 1 caption for further details.

| year | n | d | $D^h$ | $D^z$ | $\sigma_b^2$ | $\sigma_o^2$ |
|------|-----|----|----|-----|------|------|
| 1960 | 396 | 55 | 50 | 217 | 4.42 | 2.21 |
| 1970 | 670 | 42 | 53 | 222 | 3.80 | 1.90 |
| 1980 | 615 | 45 | 62 | 211 | 4.58 | 2.29 |
| 1990 | 560 | 47 | 52 | 210 | 2.88 | 1.44 |
| 2000 | 596 | 46 | 51 | 210 | 2.99 | 1.49 |
| 2010 | 637 | 45 | 64 | 212 | 4.70 | 2.35 |