# Peer review of "seNorge\_2018, daily precipitation and temperature datasets over Norway"

_Earth System Science Data, 2019_

## Referee Comment (RC1) · Anonymous Referee #1 · 14 May 2019

The authors present a new version of the seNorge dataset which is a gridded dataset for precip and temperature based on observational data. The resolution of the dataset is high with 1 km - especially considering the challenges the complex terrain poses to this effort. A state-of-the-art scale-separation approach is used for gridding which adds detail with subsequent additions of maps with even finer scales. Uncertainty estimates are made and uncertainty is related to station density.

The study is interesting and relevant and the manuscript fits within the scope of the journal. The group of authors are leading in this field and approach in the paper can serve as a guide for other NMHSs to provide gridded dataset. However, the main concerns relate to the readability of the manuscript. There will be specialists among the readers of ESSD who might disagree with this concern, but to have an impact which

is beyond the small group of specialists in advanced geostatistical techniques, a little more background and explanation is required. In addition, there are some issues on the approach that need further clarification.

Main concerns

*) The story line is sometimes hard to follow and the readability would improve with some more illustration. For instance: a brief introduction to optimal interpolation could be helpful on page 5. Another example are the scales you introduce on page 9 (line 25-27). A visualisation of some of these scales would help in guiding the reader to understand the approach. Similarly on page 10, line 15-16: can you visualize this scale somehow and show how this scale changes over time?

Figure 1: I understand that the explanation of the colour coding in fig 1a and 1b is complex, but now the reader has to read a substantial part of the article first before he/she understands what you are plotting here. A intuitive explanation for IDI might help for the reader who has a look at the figures first before deciding to read the paper. Explanation of abbreviations IDI and CV-IDI also helps. In the precipitation plot I'm missing the station locations.

*) A smaller issue is the structure of the text, a critical look would help here. For instance, on page 3, line 10 you start to claim that your approach will capture field variability at unresolved spatial scales. The next line is not an explanation how this is achieved, but deals with something quite different. It is untill line 17 that the reader is informed how you take-in the information on the unresolved spatial scale. Another example is on page 5, line 30. You write '...and Fig 1 shows those regions'. It would help if you guide the reader more explicitly where to look in Fig 1 (which regions/colours, which subfigure).

*) Relating to the interpretation of the results: Figure 2 shows that the analysis of TN is increasing with increasing CV-IDI Both the background and CV-analysis are decreasing with CV-IDI, but for summer this is not seen in the analysis. In winter this effect is absent

as well. My first guess was that this might be the influence of the urban stations, picking up the urban heat island effect. Can you comment on this?

*) In addition: In the introduction there is a paragraph concerning the effective resolution of grids. This makes me curious about the effective grid resolution of seNorge_2018 compared to seNorge2. Is there a way to quantify this? This is an interesting aspect, since the number of station observations (∼density of network) is similar in both datasets.

Other points the authors may want to look into

*) On page 4 (bottom) you describe the increase in station density and that many of these stations are installed in cities and villages. I was wondering if this aspect would give you a possibility to assess the Urban Heat island effect in Norway's larger cities? A comment on this would be interesting.

*) Page 5, line 25. Wouldn't the complexity of the topography be a relevant function here, and if so, have you looked into topography complexity? (slope, aspect, elevation)

*) Page 7, line 4-5. It is a good thing that physical consistency is enforced. A brief explanation how this is done is helpful, i.e. are you simply setting tn to tg where you find that tn is larger than tg, or is there a slightly more sophisticated approach?

*) On page 8, line 4: what is the motivation to choose a 50x50 grid? Where there any sensitivity analysis to support the choice?

*) The usage of 100 scales with a minimum of 2 and a maximum of 1400 km is unclear to the reviewers. Can you visualize some of these scales? Is there a more graphical way to explain how you use these successive scales for downscaling. Related to this, can you visualize the critical scale mentioned on page 10, line 15? Such a graphical representation convinces readers about the innovation of your method.

*) page 11, line 24: can you comment if you think there are other ways that might alleviate this problem with TN without having to install new stations?

*) page 12, line 5: here you claim that the addition of the land area fraction in equation 7 improves the temperature fields. What you are showing is the difference. Intuitively I see where you are going, but showing an improvement requires the cross validation, and the reviewer has not seen evidence that the new dataset improves considerably along the coastline.

*) page 12, line 31: I agree with your statement, but the reverse does not seem to be true. In the Oslo fjord the station density is very high but the TN quality is as low as in less dense regions.

*) The results section starts with the description of the CV. There are some concerns about the validation methods. A LOOCV or random sampling with k=folds does assume data points are spatially independent. This does not hold for data dense regions, moreover these data points will be predicted accurately due to their spatial dependence (especially using LOOCV). It is expected that the current approach will result in an underestimation of predictions errors. A way of making the validation procedure less spatially dependent (and less computationally expensive) could be to splits the datapoint into k-equal area folds.

*) page 6, line 5,8: What is the motivation to choose the numerical boundaries for CV-IDI to indicate data dense and sparse regions?

*) page 10, line 19: This is not a general description of CV but LOOCV.

*) General remark on figures: in most of the figures I'm missing the subfigure annotations (a,b,c). Please include a raster grid with lat/lon for the spatial plots. For regularly gridded raster plots one scale bar is sufficient.

*) Figure 4: Good to include lat/lon averaged differences, am I assuming correctly that the grey area are the min/max values (between -4 and +1.5)? The lateral and bottom panels y-axis temperatures are hard to read. This does not support the text on page 11, line 28, which suggests that almost all differences are between -2 and +1.

*) page 7, line 18: what does the i mean in the definition of G and S as these latter quantities appear not to be related to grid point i?

*) Table 1: The caption states that the third station is used while in the text (page 8, line 33) the average distance to the nearest four stations is used.

Very minor issues

*) page 3, line 27: "Finally, Section 4..." → This suggests there is no chapter 5.

*) page 7, line 18: typo in gridpoint

*) page 10, line 7: I assume "to have the same error"

*) page 13, line 8: Shouldn't 0.4 be 40%?

*) page 13, line 18: I guess this should be "paper of Lussana"

*) page 17, missing pages in the Reistad citation

*) Figure 7: Using black in the color scale is inconvenient since country borders and coastlines are also black.

---

## Author Comment (AC1) · 6 Jun 2019

Dear Reviewer,

We would like to thank the reviewer for the precious comments that will help us in improving the quality and readability of our manuscript. We are deeply grateful for that.

In general, We will revise the whole manuscript so to improve its readability. In particular, the issues raised by the reviewer will be addressed. The additional work we planned onto the manuscript will include:

- The quantification of the effective resolution for the precipitation field and its comparison with seNorge2
- A step-by-step graphical representation of the precipitation spatial interpolation algorithm
- The quantification of the improvements due to the introduction of land area fraction in the analysis, as compared to seNorge2 (which does not use land area fraction)

The answers to the comments follow. The reviewer's comments are reported in blue. For brevity's sake, missing answers mean that we will adjust the text as suggested by the reviewer.

Main concerns

*) The story line is sometimes hard to follow and the readability would improve with some more illustrations. For instance: a brief introduction to optimal interpolation could be helpful on page 5. Another example are the scales you introduce on page 9 (line25-27). A visualisation of some of these scales would help in guiding the reader to understand the approach. Similarly on page 10, line 15-16: can you visualize this scale somehow and show how this scale changes over time?

We will extend the description of OI. As for the scales, we will include a dedicated Figure.

Figure 1: I understand that the explanation of the colour coding in fig 1a and 1b is complex, but now the reader has to read a substantial part of the article first before he/she understands what you are plotting here. An intuitive explanation for IDI might help for the reader who has a look at the figures first before deciding to read the paper. Explanation of abbreviations IDI and CV-IDI also helps. In the precipitation plot I'm missing the station locations.

In the precipitation plot, stations are located in the middle of beige areas. We will draw the points marking the station locations.

*) A smaller issue is the structure of the text, a critical look would help here. For Instance, on page 3, line 10 you start to claim that your approach will capture field variability at unresolved spatial scales. The next line is not an explanation of how this is achieved, but deals with something quite different. It is until line 17 that the reader is informed how you take-in the information on the unresolved spatial scale. Another example is on page 5, line 30. You write '...and Fig 1 shows those regions'. It would help if you guide the reader more explicitly where to look in Fig 1 (which regions/colours,which subfigure).

*) Relating to the interpretation of the results: Figure 2 shows that the analysis of TN is increasing with increasing CV-IDI Both the background and CV-analysis are decreasing with CV-IDI, but for summer this is not seen in the analysis. In winter this effect is absent as well. My first guess was that this might be the influence of the urban stations, picking up the urban heat island effect. Can you comment on this?

Preliminary notes:
1. At an arbitrary location, CV-analysis and background are -by construction-independent of the observed value. On the other hand, the observed value is used in the calculation of the analysis.
2. OI provides the best (i.e. minimum analysis error variance) linear unbiased estimate of the unknown true value of a quantity. The analysis is the expected value of such an estimate. In practice:
    a. In data-dense areas: the analysis at an arbitrary location is influenced by several nearby observations, such that the analysis is always a bit different from the observation measured at that location.
    b. In data-sparse areas: the analysis at an arbitrary location is much more influenced by the associated observation than by other observations, as a consequence the analysis tends to stay closer to that observation.

Points 1-2 explain why the CV-analysis and background performances increase with the increasing of the station density, while the analysis performance decreases (or stay constant) with the increase in station density.

As the reviewer pointed out, the TN analysis behaves quite differently from TG and TX analyses. Our explanation is that the procedure used for the construction of the pseudo-background provides less satisfactory results for TN than for TG and TX. For TN, on average the background differs from the observations more than for TG and TX. The OI realizes a trade-off between observations and background and a worse background causes inevitably a more uncertain OI estimate. However, in data-dense

regions the observations "cooperate" to pull the analyses towards the observations. As a consequence, TN analyses score better for data-dense than for data-sparse regions. In conclusion, we think that the current pseudo-background is more suited for TG and TX than for TN. We will emphasize this conclusion in the manuscript. It is worth remarking that despite the increased uncertainty, the TN fields do provide valuable information for the computation of e.g. climate indices.

*) In addition: In the introduction there is a paragraph concerning the effective resolution of grids. This makes me curious about the effective grid resolution of seNorge_2018 compared to seNorge2. Is there a way to quantify this? This is an interesting aspect, since the number of station observations (~density of network) is similar in both datasets.

The quantification of the effective resolution of meteorological gridded fields is a fascinating and challenging problem. We will apply a scale-decomposition approach (based on 2D wavelet) to the daily precipitation fields of seNorge_2018 and seNorge2. Then, we will study the differences in terms of aggregated statistics for each scale. At the moment, we don't plan to analyze the effective resolution of temperature fields.

Other points the authors may want to look into:
*) On page 4 (bottom) you describe the increase in station density and that many of these stations are installed in cities and villages. I was wondering if this aspect would give you a possibility to assess the Urban Heat island effect in Norway's larger cities? A comment on this would be interesting.

The norwegian cities, even the largest ones, are surrounded by forests, they include large green areas, are usually orographically complex and they are invariably close to the sea. Such a heterogeneity of land uses and the alternation of land and water possibly makes the accurate assessment of UHI effects more difficult than elsewhere. Our guess is that a much denser station network would be needed to properly represent the UHI over Norwegian cities. Ideally, at least one observation point for each significant change in land use should be present and this is not always the case for our network. However, our network is dense enough to assess the impact of cities and study the differences between urban areas and forests at a regional level.

*) Page 5, line 25. Wouldn't the complexity of the topography be a relevant function here, and if so, have you looked into topography complexity? (slope, aspect, elevation)

No, we haven't looked into that. In the first place, the successful application of a statistical model based on several geographical parameter for precipitation (on a daily time scale) implicitly assumes the local availability of several stations (e.g, one for each slope/aspect/elevation classes in a valley) and this is often not the case for the Norwegian network, as for many other networks in mountainous regions. An interesting analysis on this point has been made by Masson and Frei (2014), where they pointed out that "Our results confirm that the consideration of topography effects is important for spatial interpolation of precipitation in high-mountain regions. But a single predictor may be sufficient and taking appropriate account of the spatial autocorrelation (by kriging) can be more effective than the development of elaborate predictor sets within a regression model."

Masson, D. and Frei, C.: Spatial analysis of precipitation in a high-mountain region: exploring methods with multi-scale topographic predictors and circulation types, Hydrol. Earth Syst. Sci., 18, 4543-4563, https://doi.org/10.5194/hess-18-4543-2014, 2014.

*) Page 7, line 4-5. It is a good thing that physical consistency is enforced. A brief explanation of how this is done is helpful, i.e. are you simply setting tn to tg where you find that tn is larger than tg, or is there a slightly more sophisticated approach?

We have implemented a brute force approach that will be described in the text. It is not that different from the simple method described by the reviewer.

*) On page 8, line 4: what is the motivation to choose a 50x50 grid? Where there any sensitivity analysis to support the choice?

We will include the motivations in the text. Ideally, one may estimate a pseudo-background field centered on each gridpoint of the 1km grid. To speed up the elaboration the background is computed on a 50x50 grid, then "downscaled" onto the 1km grid. In our case, the average distance between nearby points used as centroids for the background calculation (i.e., centers of each grid cell, 50x50 grid) is 27.5 km (averaged width and height, see page 8 line 18), which is less than the average distance between a station and its 3-4 closest ones. This way, we are sure to represent in the final background field all the relevant features observed by the network.

*) The usage of 100 scales with a minimum of 2 and a maximum of 1400 km is unclear to the reviewers. Can you visualize some of these scales? Is there a more graphical way to explain how you use these successive scales for downscaling. Related to this, can you visualize the critical scale mentioned on page 10, line 15?

Such a graphical representation convinces readers about the innovation of your method.

We will introduce a graphical step-by-step description of the interpolation method for precipitation so to make the whole discussion about the spatial scale less abstract.

*) page 11, line 24: can you comment if you think there are other ways that might alleviate this problem with TN without having to install new stations?

As an alternative to the installation of new stations, it would be possible to modify the procedure used to compute the background field. The problem with TN seems to be present both on the regional scale and on the local scale. A better background will improve at least the representation at the regional scale. The use of numerical model output fields (e.g., reanalysis) could also allow us to improve the large scale TN.

*) page 12, line 5: here you claim that the addition of the land area fraction in equation 7 improves the temperature fields. What you are showing is the difference. Intuitively I see where you are going, but showing an improvement requires the cross validation, and the reviewer has not seen evidence that the new dataset improves considerably along the coastline.

The reviewer remark is correct. We need to show that improvement. We will include in the text the verification based on cross-validation and considering coastal stations only.

*) page 12, line 31: I agree with your statement, but the reverse does not seem to be true. In the Oslo fjord the station density is very high but the TN quality is as low as in less dense regions.

Fig. 7, TN DJF. The graph in the box (% of large errors vs IDI) shows that TN in data dense region is less likely to provide large errors than in data sparse regions. The color scale in the TN map is the same as for TG and TN so to show the larger uncertainties associated with TN.

*) The results section starts with the description of the CV. There are some concerns about the validation methods. A LOOCV or random sampling with k=folds does assume data points are spatially independent. This does not hold for data dense regions, moreover these data points will be predicted accurately due to their spatial dependence (especially using LOOCV). It is expected that the current approach will result in an underestimation of prediction errors. A way of making the validation

procedure less spatially dependent (and less computationally expensive) could be to split the data point into k-equal area folds.

Our description of the CV procedure used for precipitation is incomplete, since for precipitation we designed the procedure used for the random selection of stations such that the chosen stations are not too close to each other (i.e., approximately equidistant). We will explain it better in the text.
In general, it is exactly to avoid underestimating the prediction errors that we have introduced the discussion on station density and analysis errors, that bring us to Figs. 2-3-5-7. Prediction errors (or analysis uncertainties) do vary over the domain, mostly depending on station density, and this aspect is taken into account in our study.

*) page 6, line 5,8: What is the motivation to choose the numerical boundaries for CV-IDI to indicate data dense and sparse regions?

At a specific location (and with the intention to use it as a surrogate for a gridpoint): a value of CV-IDI=0 means that the observations do not have any influence at all on the CVanalysis; CV-IDI=1 means that the observations will strongly condition the CVanalysis, no matter the background value. The numerical boundaries are somewhat arbitrarily chosen: with CV-IDI values > 0.85 we are sure that the observations will strongly influence the CVanalysis; CV-IDI<0.45 means that the observations will play a minor role in the CVanalysis, if compared to the background. The two intermediate classes have been chosen so to ensure having some observations in each class.

*) page 10, line 19: This is not a general description of CV but LOOCV.

We will modify the text.

*) General remark on figures: in most of the figures I'm missing the subfigure annotations (a,b,c). Please include a raster grid with lat/lon for the spatial plots. For regularly gridded raster plots one scale bar is sufficient.

*) Figure 4: Good to include lat/lon averaged differences, am I assuming correctly that the grey area are the min/max values (between -4 and +1.5)? The lateral and bottom panels y-axis temperatures are hard to read. This does not support the text on page11, line 28, which suggests that almost all differences are between -2 and +1.

Yes, correct. We will modify the text accordingly.

*) page 7, line 18: what does the i mean in the definition of G and S as these latter quantities appear not to be related to grid point i?

G and S are related to the i-th gridpoint because for each gridpoint a different horizontal decorrelation length is used.

*) Table 1: The caption states that the third station is used while in the text (page 8, line33) the average distance to the nearest four stations is used.

Very minor issues

*) page 3, line 27: "Finally, Section 4..."→This suggests there is no chapter 5.

*) page 7, line 18: typo in gridpoint

*) page 10, line 7: I assume "to have the same error"

*) page 13, line 8: Shouldn't 0.4 be 40%?

ETS is normally indicated as a number between -⅓ and 1 (https://www.wmo.int/pages/prog/arep/wwrp/new/jwgfvr.html).

*) page 13, line 18: I guess this should be "paper of Lussana"

*) page 17, missing pages in the Reistad citation

*) Figure 7: Using black in the color scale is inconvenient since country borders and coastlines are also black.

---

## Referee Comment (RC2) · Anonymous Referee #2 · 4 Jul 2019

**Review of "seNorge_2018, daily precipitation and temperature datasets over Norway" By Cristian Lussana et al.**

The authors present a new version of Norwegian national daily temperature and precipitation interpolated daily fields from the latter half of the 20th Century to date. The data product shall, undoubtedly, constitute a valuable national resource for decision makers within Norway and the broader Nordic region. To build confidence in the product peer review is certainly a necessary condition and thus I see eventual publication as important. In reviewing the discussion paper there are a number of issues that I believe the authors must address prior to acceptance.

Firstly, the paper structure requires significant work. The introduction mixes methods and discussion. The data section has a significant amount of methods in it and does not clearly denote the various observational / model sources used. Presently there is very limited description / analysis of the derived spatial fields and characterisation which would be important for users. Finally, there is no discussion section. My suggestion would be to substantively restructure the paper for readability into sections that go:
- introduction,
- data,
- methods,
- product analysis (including showing some example applications),
- verification,
- discussion and
- conclusion.

Then significant effort is required to shuffle content around to fit that structure, ensuring that relevant text ends up in the appropriate section. Given the need to restructure the paper I shall not point out minor typographical issues in the expectation that they may not persist under any revised structure. A number of other minor points should also, naturally, be resolved by undertaking such a restructure so I do not make these further here. Such a restructuring to my view is essential prior to acceptance.

In terms of the methods there is a significant issue in offsetting Tx/Tn from Tg by 12 hours. If Tg is the mean of 06 to 06 but Tx and Tn are maximum and minimum between 18 and 18 it is physically impossible to robustly assess consistency. This has been shown in e.g. GHCND and can follow from several toy examples you may wish to play with whereby for example a very strong warm front passes through at midnight which would be seen in one day for Tx and Tn but another day for Tg and may lead to an over-propensity of flagging good data as dubious accordingly. This propensity will vary seasonally (higher in winter half year) and geographically (higher further north / inland) where both diurnal structure decreases and synoptic variability increases. Significant justification would be required for maintaining the use of days offset by 12 hours for the three temperature elements and my strong recommendation would be to align these to the same time which would greatly simplify the analysis and assure better geophysical consistency with fewer false flags. It would also aid usability considerably to align the times for all 4 elements. So, whether you choose 06 to 06, 18 to 18 or some other times I would very strongly urge aligning the times used to define the day here which would enable greater usability and improved cross-checking.

The authors make a throw away remark at the end of page 2 regarding suitability for long-term trend characterisation which seems to rely upon findings of a prior analysis. It is unclear whether the findings would persist into the present dataset in the Norwegian context. It is necessary, in my view, to show this and the suggested change in overall paper structure should facilitate this.

In the methods $X_i$ is used twice, one should be $X_j$. Then the same overhat nomenclature is used to denote both a point estimate and a spatial scale. This is very confusing to the reader in what is already a very statistically dense paper. Assuming that the average ESSD paper is not a statistician it would be very useful to simplify where possible the discussion of methods and certainly to use unique notations when talking about distinct things so as to not confuse unnecessarily your readers. Overall, a reduction in the number of equations would likely serve the ESSD readership.

The right hand panel of figure 6 uses a non-intuitive colour scale whereby wetter values are red and drier values blue (as I understand this panel at least). If I am correct it would be advisable to flip the colour bar so that the colours intuitively map to wet / dry rather than doing so counter-intuitively. If I am wrong then an improved explanation is required.

---

## Author Comment (AC2) · 15 Jul 2019

Dear Reviewer,

We would like to thank you for the comments that will help us in improving the quality and readability of our manuscript. We are deeply grateful for that.

Point-by-point answers to the comments follow. The reviewer's letter is reported in Italic.

*The authors present a new version of Norwegian national daily temperature and precipitation interpolated daily fields from the latter half of the 20thCentury to date. The data product shall, undoubtedly, constitute a valuable national resource for decision makers within Norway and the broader Nordic region. To build confidence in the product peer*

[Figure]

*review is certainly a necessary condition and thus I see eventual publication as important. In reviewing the discussion paper there are a number of issues thatI believe the authors must address prior to acceptance.*

*Firstly, the paper structure requires significant work. The introduction mixes methods and discussion. The data section has a significant amount of methods in it and does not clearly denote the various observational / model sources used. Presently there is very limited description / analysis of the derived spatial fields and characterisation which would be important for users. Finally, there is no discussion section. My suggestion would be to substantively restructure the paper for readability into sections that go:*

- *introduction,*

- *data,*

- *methods,*

- *product analysis(including showing some example applications),*

- *verification,*

- *discussion and*

- *conclusion.*

*Then significant effort is required to shuffle content around to fit that structure, ensuring that relevant text ends up in the appropriate section.Given the need to restructure the paper I shall not point out minor typographical issues in the expectation that they may not persist under any revised structure.A number of other minor points should also, naturally, be resolved by undertaking such a restructure so I do not make these further here.Such a restructuring to my view is essential prior to acceptance.*

Reply: we will modify the structure of the paper as suggested.

*In terms of the methods there is a significant issue in offsetting Tx/Tn from Tg by 12 hours. If Tg is the mean of 06 to 06 but Tx and Tn are maximum and minimum between 18 and 18 it is physically impossible to robustly assess consistency. This has been shown in e.g. GHCND and can follow from several toy examples you may wish to play with whereby for example a very strong warm front passes through at midnight which would be seen in one day for Tx and Tn but another day for Tg and may lead to an over-propensity of flagging good data as dubious accordingly.This propensity will vary seasonally (higher in winter half year) and geographically (higher further north/ inland) where both diurnal structure decreases and synoptic variability increases.Significant justification would be required for maintaining the use of days offset by 12 hours for the three temperature elements and my strong recommendation would be to align these to the same time which would greatly simplify the analysis and assure better geophysical consistency with fewer false flags.It would also aid usability considerably to align the times for all 4 elements.So, whether you choose 06 to 06, 18 to 18 or some other times I would very strongly urge aligning the times used to define the day here which would enable greater usability and improved cross-checking.*

Reply: We will clarify in the text the reasons why we have to use two different definitions of day. As stated at page 4 "TG and RR share the same day-definition so as to better serve hydrological applications, while for historical reasons TX and TN have a different day definition." What we mean by this statement is that the historical measurements back to 1957 have been performed with different offsets. Since our gridded datasets are based on observations, we are forced to use the same day-definition as the observed data. We agree with the reviewer that the ideal situation would be to have the 4 variables aligned on the same day definition. Due to the lack of data, this is not possible. On the one hand, this lack of consistency may constitute a problem for some users. On the other hand, our experience is that TX, TN, RR and TG do provide useful information in numerous applications both for monitoring the ongoing weather and for research. As an example, we may mention two applications that are particularly important for the Norwegian society: (1) hydrological numerical models operated by the national civil protection authorities make profitable use of seNorge_2018 TG and RR; (2) TX and TN can be used to compute e.g., widely used climate indices (http://surfobs.climate.copernicus.eu/userguidance/indicesdictionary.php).

*The authors make a throw away remark at the end of page 2 regarding suitability for long-term trend characterisation which seems to rely upon findings of a prior analysis. It is unclear whether the findings would persist into the present dataset in the Norwegian context. It is necessary, in my view, to show this and the suggested change in overall paper structure should facilitate this.*

Reply: we will clarify this important point in the revised paper. The users must be aware that variations in the observational network do have an impact on the gridded datasets.

*In the methods Xi is used twice, one should be Xj. Then the same overhat nomenclature is used to denote both a point estimate and a spatial scale. This is very confusing to the reader in what is already a very statistically dense paper. Assuming that the average ESSD paper is not a statistician it would be very useful to simplify where possible the discussion of methods and certainly to use unique notations when talking about distinct things so as to not confuse unnecessarily your readers. Overall, a reduction in the number of equations would likely serve the ESSD readership.*

Reply: We will revise the description of the method as suggested by the reviewer. It is worth remarking that the mathematical notation adopted is based on Ide et al. (1997), that is widely used in data assimilation (DA) and described in several books (e.g., Kalnay 2003). By adopting those standards, readers that are familiar with DA (not necessarily statisticians) will recognize the equations used and, at the same time, those readers that are not familiar with DA can rely on the vast literature on this topic.

References: Ide, K., Courtier, P., Ghil, M. and Lorenc, A.C., 1997. Unified Notation for Data Assimilation: Operational, Sequential and Variational (Special Issue Data Assimilation in Meteorology and Oceanography: Theory and Practice). Journal of the Meteorological Society of Japan. Ser. II, 75(1B), pp.181-189.

Kalnay, E., 2003. Atmospheric modeling, data assimilation and predictability. Cambridge university press.

*The right hand panel of figure 6 uses a non-intuitive colour scale whereby wetter values are red and drier values blue(as I understand this panel at least). If I am correct it would be advisable to flip the colour bar so that the colours intuitively map to wet / dry rather than doing so counter-intuitively. If I am wrong then an improved explanation is required.*

Reply: we will modify the figure as suggested.

---

## Author Response (AR1)

Dear Reviewers,

Once again, thanks for your comments and suggestions. You may have noticed that we have answered to your comments in the interactive discussion and for your convenience we copy here the link:
https://www.earth-syst-sci-data-discuss.net/essd-2019-43/

We have modified the structure of the paper, according to your suggestions. The revised manuscript is organized as follows:

- Introduction: in order to describe the original aspects of our research, we need to introduce concepts such as the effective resolution of a dataset and the definitions of spatial scales (in a way that is useful for our purposes). Those paragraphs may also be regarded as methods, nonetheless we believe they belong in the introduction.
- Data: this Section has been split into three sub-sections. First, the observations are described, together with the procedure used to post-process precipitation measured data. Second, we describe the reference fields for the spatial interpolation of precipitation. Third, IDI is introduced and discussed here, because we intend
- Methods: the core of our paper in the spatial analysis. For this reason, we believe the methods should be focused on spatial interpolation (e.g., post-processing of precipitation measurements is part of the Data section).
- Example application for precipitation: new section. The spatial analysis of precipitation is here described step-by-step. As stated in the last paragraph of the Introduction, examples of temperature analyses for individual cases can be found in a previous paper. We have put a lot of effort in the characterization of the analysis products in the verification section.
- Verification: the name of this section has been changed from Results to Verification. The text has been revised and the section has been split into Verification and discussion. We have added the paragraph on Precipitation and effective resolution.
- Discussion: new section. Some important issues have been discussed in detail, such as the TG, TX and TN cross-checking.
- Conclusions.

Additional calculations have been performed, to either support our conclusions or point out new features of seNorge_2018. In particular: the effective resolution of RR has been assessed; the significance of land-area-fraction in reducing the uncertainties in the analysis of coastal stations has been verified through cross-validation.

Several figures have been added and most of the original figures have been modified.

Point-by-point response to your reviews follows. The Reviewers' comments are reported in Italic.

Best Regards,
Cristian Lussana on behalf of the Authors

**Authors' Response to Reviewer 1**

*Main concerns*

*\*) The story line is sometimes hard to follow and the readability would improve with some more illustrations. For instance: a brief introduction to optimal interpolation could be helpful on page 5. Another example are the scales you introduce on page 9 (line25-27). A visualisation of some of these scales would help in guiding the reader to understand the approach. Similarly on page 10, line 15-16: can you visualize this scale somehow and show how this scale changes over time?*

**Reply:** We have completely revised the structure of the manuscript. We have included an example of application for RR and the figure mentioned in your comment has been placed there.

*Figure 1: I understand that the explanation of the colour coding in fig 1a and 1b is complex, but now the reader has to read a substantial part of the article first before he/she understands what you are plotting here. An intuitive explanation for IDI might help for the reader who has a look at the figures first before deciding to read the paper. Explanation of abbreviations IDI and CV-IDI also helps. In the precipitation plot I'm missing the station locations.*

**Reply:** Fig. 1 has been modified such that now the issues raised by the Reviewer should be solved.

*\*) A smaller issue is the structure of the text, a critical look would help here. For Instance, on page 3, line 10 you start to claim that your approach will capture field variability at unresolved spatial scales. The next line is not an explanation of how this is achieved, but deals with something quite different. It is until line 17 that the reader is informed how you take-in the information on the unresolved spatial scale. Another example is on page 5, line 30. You write '...and Fig 1 shows those regions'. It would help if you guide the reader more explicitly where to look in Fig 1 (which regions/colours,which subfigure).*
**Reply:** we have revised the manuscript as suggested.

*\*) Relating to the interpretation of the results: Figure 2 shows that the analysis of TN is increasing with increasing CV-IDI Both the background and CV-analysis are decreasing with CV-IDI, but for summer this is not seen in the analysis. In winter this effect is absent as well. My first guess was that this might be the influence of the urban stations, picking up the urban heat island effect. Can you comment on this?*

**Reply:** with reference to the interactive comments, we have included a comment on this topic in the manuscript (Section "Discussion").

*\*) In addition: In the introduction there is a paragraph concerning the effective resolution of grids. This makes me curious about the effective grid resolution of seNorge_2018 compared to seNorge2. Is there a way to quantify this? This is an interesting aspect, since the number of station observations (~density of network) is similar in both datasets.*

**Reply:** We have used a scale-decomposition approach (based on 2D wavelet) to the daily precipitation fields of seNorge_2018 and seNorge2. The results have been included in the Section "Discussion", together with a dedicated Figure.

*Other points the authors may want to look into:*
*\*) On page 4 (bottom) you describe the increase in station density and that many of these stations are installed in cities and villages. I was wondering if this aspect would give you a possibility to assess the Urban Heat island effect in Norway's larger cities? A comment on this would be interesting.*

**Reply:** see the interactive comments.

*\*) Page 5, line 25. Wouldn't the complexity of the topography be a relevant function here, and if so, have you looked into topography complexity? (slope, aspect, elevation)*

**Reply:** see the interactive comments.

*\*) Page 7, line 4-5. It is a good thing that physical consistency is enforced. A brief explanation of how this is done is helpful, i.e. are you simply setting tn to tg where you find that tn is larger than tg, or is there a slightly more sophisticated approach?*

**Reply:** We have added some text to the manuscript to describe the check.

*\*) On page 8, line 4: what is the motivation to choose a 50x50 grid? Where there any sensitivity analysis to support the choice?*

**Reply:** We have added some text in the Section "Methods" of the manuscript to better describe why we have chosen a 50x50 grid.

*\*) The usage of 100 scales with a minimum of 2 and a maximum of 1400 km is unclear to the reviewers. Can you visualize some of these scales? Is there a more graphical way to explain how you use these successive scales for downscaling.*

*Related to this, can you visualize the critical scale mentioned on page 10, line 15? Such a graphical representation convinces readers about the innovation of your method.*

**Reply:** A graphical representation has been added.

*\*) page 11, line 24: can you comment if you think there are other ways that might alleviate this problem with TN without having to install new stations?*

**Reply:** see the interactive comments. We have added a discussion on this point in the text.

*\*) page 12, line 5: here you claim that the addition of the land area fraction in equation 7 improves the temperature fields. What you are showing is the difference. Intuitively I see where you are going, but showing an improvement requires the cross validation, and the reviewer has not seen evidence that the new dataset improves considerably along the coastline.*

**Reply:** We have included in the text the verification based on cross-validation and considering coastal stations only.

*\*) page 12, line 31: I agree with your statement, but the reverse does not seem to be true. In the Oslo fjord the station density is very high but the TN quality is as low as in less dense regions.*

**Reply:** see the interactive comments.

*\*) The results section starts with the description of the CV. There are some concerns about the validation methods. A LOOCV or random sampling with k=folds does assume data points are spatially independent. This does not hold for data dense regions, moreover these data points will be predicted accurately due to their spatial dependence (especially using LOOCV). It is expected that the current approach will result in an underestimation of prediction errors. A way of making the validation procedure less spatially dependent (and less computationally expensive) could be to split the data point into k-equal area folds.*

**Reply:** see the interactive comments. We have modified the text so to take into account the Reviewer's remark.

*\*) page 6, line 5,8: What is the motivation to choose the numerical boundaries for CV-IDI to indicate data dense and sparse regions?*

**Reply:** we have included an explanation in the manuscript.

*) page 10, line 19: This is not a general description of CV but LOOCV.

**Reply:** The text has been modified.

*) General remark on figures: in most of the figures I'm missing the subfigure annotations (a,b,c). Please include a raster grid with lat/lon for the spatial plots. For regularly gridded raster plots one scale bar is sufficient.

**Reply:**

*) Figure 4: Good to include lat/lon averaged differences, am I assuming correctly that the grey area are the min/max values (between -4 and +1.5)? The lateral and bottom panels y-axis temperatures are hard to read. This does not support the text on page11, line 28, which suggests that almost all differences are between -2 and +1.

**Reply:** We have modified the text accordingly.

*) page 7, line 18: what does the i mean in the definition of G and S as these latter quantities appear not to be related to grid point i?

**Reply:** G and S are related to the i-th gridpoint because for each gridpoint a different horizontal decorrelation length is used.

*) Table 1: The caption states that the third station is used while in the text (page 8, line33) the average distance to the nearest four stations is used.

**Reply:** we have modified the text so to correct for the mistake.

*Very minor issues*

**Reply:** thanks for pointing out the following issues.

*) page 3, line 27: "Finally, Section 4..."→This suggests there is no chapter 5.

*) page 7, line 18: typo in gridpoint

*) page 10, line 7: I assume "to have the same error"

*) page 13, line 8: Shouldn't 0.4 be 40%?

*) page 13, line 18: I guess this should be "paper of Lussana"

*) page 17, missing pages in the Reistad citation

*) Figure 7: Using black in the color scale is inconvenient since country borders and coastlines are also black.

**Authors' Response to Reviewer 2**

*The authors present a new version of Norwegian national daily temperature and precipitation interpolated daily fields from the latter half of the 20thCentury to date. The data product shall, undoubtedly, constitute a valuable national resource for decision makers within Norway and the broader Nordic region. To build confidence in the product peer review is certainly a necessary condition and thus I see eventual publication as important. In reviewing the discussion paper there are a number of issues thatI believe the authors must address prior to acceptance.*

*Firstly, the paper structure requires significant work. The introduction mixes methods and discussion. The data section has a significant amount of methods in it and does not clearly denote the various observational / model sources used. Presently there is very limited description / analysis of the derived spatial fields and characterisation which would be important for users. Finally, there is no discussion section. My suggestion would be to substantively restructure the paper for readability into sections that go:*
- *introduction,*
- *data,*
- *methods,*
- *product analysis(including showing some example applications),*
- *verification,*
- *discussion and*
- *conclusion.*

*Then significant effort is required to shuffle content around to fit that structure, ensuring that relevant text ends up in the appropriate section.Given the need to restructure the paper I shall not point out minor typographical issues in the expectation that they may not persist under any revised structure.A number of other minor points should also, naturally, be resolved by undertaking such a restructure so I do not make these further here.Such a restructuring to my view is essential prior to acceptance.*

**Reply:** we have modified the structure of the paper.

*In terms of the methods there is a significant issue in offsetting Tx/Tn from Tg by 12 hours. If Tg is the mean of 06 to 06 but Tx and Tn are maximum and minimum between 18 and 18 it is physically impossible to robustly assess consistency. This has been shown in e.g. GHCND and can follow from several toy examples you may wish to play with whereby for example a very strong warm front passes through at midnight which would be seen in one day for Tx and Tn but another day for Tg and may lead to an over-propensity of flagging good data as dubious accordingly.This*

*propensity will vary seasonally (higher in winter half year) and geographically (higher further north/ inland) where both diurnal structure decreases and synoptic variability increases.Significant justification would be required for maintaining the use of days offset by 12 hours for the three temperature elements and my strong recommendation would be to align these to the same time which would greatly simplify the analysis and assure better geophysical consistency with fewer false flags.It would also aid usability considerably to align the times for all 4 elements. So, whether you choose 06 to 06, 18 to 18 or some other times I would very strongly urge aligning the times used to define the day here which would enable greater usability and improved cross-checking.*

**Reply:** We have to use two different definitions of day, as reported in the interactive discussion. This fact is stated explicitly in the definition of the variables (beginning of Section 2.1).

*The authors make a throw away remark at the end of page 2 regarding suitability for long-term trend characterisation which seems to rely upon findings of a prior analysis. It is unclear whether the findings would persist into the present dataset in the Norwegian context. It is necessary, in my view, to show this and the suggested change in overall paper structure should facilitate this.*

**Reply:** we have rephrased the statement so to avoid confusion.

*In the methods Xi is used twice, one should be Xj. Then the same overhat nomenclature is used to denote both a point estimate and a spatial scale. This is very confusing to the reader in what is already a very statistically dense paper. Assuming that the average ESSD paper is not a statistician it would be very useful to simplify where possible the discussion of methods and certainly to use unique notations when talking about distinct things so as to not confuse unnecessarily your readers. Overall, a reduction in the number of equations would likely serve the ESSD readership.*

**Reply:** We have revised a bit the notation. We believe it is not possible to reduce the number of equations without compromising the possibility to reproduce our methods.

*The right hand panel of figure 6 uses a non-intuitive colour scale whereby wetter values are red and drier values blue(as I understand this panel at least). If I am correct it would be advisable to flip the colour bar so that the colours intuitively map*

*to wet / dry rather than doing so counter-intuitively. If I am wrong then an improved explanation is required.*

**Reply:** The figure has been modified as suggested.

[revised manuscript text omitted]

---

## Referee Report (RR1)

**Review of seNorge_2018, daily precipitation and temperature datasets over Norway by Christian Lussana et al.**

The authors have taken the pair of reviews and made substantive revisions to the manuscript that overall substantively improve readability of the piece as a whole. I have a number of queries and suggestions which should be considered prior to publication.

**Major comments**

1. At page 4 line 24-25 allusion is made to site exposures. A reader requires a reference to documentation of the site exposure method and further elucidation as to how, specifically, these were used.

2. The issue in p.7 line 4 over nomenclature persists. One of the Xi should be Xj surely and there should be a jth row or column?

3. The example application for precipitation discussed in section 4 relates to vigorous large circulation cyclonic precipitation event which, naively, your method may perform best at. While it is understandable that you wish to show a high skill example it may also give a misleading impression. At a minimum this should be acknowledged. Ideally a more challenging situation such as a summer convective event should also be shown.

4. Why is the section 5.3 analysis limited to winter? Surely it would be valuable to show this for at least summer in addition, if not all seasons to build user confidence and understanding of the product strengths and limitations?

5. Generally the figure captions are too short. Often text should be moved from the main body to the caption. The figure captions need to provide all information necessary for a reader to understand and interpret the figure and oftentimes this is missing.

**Minor comments**

1. Given the caveats rightly stated in the discussion the abstract at lines 11-12 on page 1 seems a bit unduly definitive?

2. P.3 line 21 described and discusses -> described and discussed

3. P.4 line 13 so to -> so as to

4. P.4 line 20 as precipitation -> as the precipitation

5. P.6 line 14 tend -> Tends?

6. Much of the paragraph starting p.6 line 28 should be moved to the figure caption rather than the text. Also: IDI equals to -> IDI equal to

7. I would provide a very brief synopsis of the section 6 discussion at p.7 line 15-16

8. P.9 line 31 so to transform -> either to transform OR so as to transform

9. P.13 lines. 18-20 should be in an expanded figure caption instead of the main text.

10. Please rephrase p.13 line 24 – it should not be about belief.

11. P.15 line 4 In this paragraph -> Next

12. Figure 3 caption projection of the differences is, I assume, an error, but I'm not sure what you actually intend to state here.

13. Figure 4 caption. Is the colour bar dimensionless units? If so state so. If not then give the units. Regardless please clarify the caption.

14. Figure 5 caption. Please help the reader out here. Are larger or smaller scales preferable? How does this scale partition between the methodological aspects and the event specific aspects? The event in question is a large cyclonic event which may have broad spatial scales. To what extent are the scales shown here a result of the event specific nature instead of the method? Intuitively the two must be intertwined and may be remedied by showing e.g. a summer convective event as suggested in major comments.

15. Figure 8 caption should end: See Fig 7 caption for further details.

16. Figure 13 lower right panel RR -> RR DJF for consistency with remaining panels

---

## Author Response (AR2)

**Answer to**
**Review of seNorge_2018, daily precipitation and temperature datasets over Norway by Cristian Lussana et al.**
* * *
* * *
Dear Reviewer,

Thanks for your thoughtful and thorough work in revising our manuscript.

Point-by-point response to your review follows. The Reviewer's comments are reported in Italic.

Best Regards,
Cristian Lussana on behalf of the Authors
* * *
* * *
*The authors have taken the pair of reviews and made substantive revisions to the manuscript that overall substantively improve readability of the piece as a whole. I have a number of queries and suggestions which should be considered prior to publication.*

*Major comments*

*1. At page 4 line 24-25 allusion is made to site exposures. A reader requires a reference to documentation of the site exposure method and further elucidation as to how, specifically, these were used.*

**Reply:** page 4 line 24, the references for seNorge version 1.1 are reported (Mohr, 2008, 2009). The references include the definition of site exposure and its use for the production of seNorge 1.1.

*2. The issue in p.7 line 4 over nomenclature persists. One of the Xi should be Xj surely and there should be a jth row or column?*

**Reply:** the mathematical notation used is correct and consistent. In the paper by Sakov and Bertino (2011) pag. 227 first column, they define the notation that we have used. However, we have modified the text such that:

- **X** (bold capital X) indicates a matrix
- $\mathbf{X}_j$ (bold capital X with "j" as subscript) indicates the j-th column of matrix **X**
- $\mathbf{X}_{i,:}$ (bold capital X with "i,:" as subscript) indicates the i-th row of matrix **X**
- $\mathbf{X}_{ij}$ (bold capital X with "ij" as subscript) indicates the element at the i-th row and j-th column of matrix **X**

*3. The example application for precipitation discussed in section 4 relates to vigorous large circulation cyclonic precipitation event which, naively, your method may perform best at. While it is understandable that you wish to show a high skill example it may also give a misleading impression. At a minimum this should be acknowledged. Ideally a more challenging situation such as a summer convective event should also be shown.*

**Reply:** The objective of the example application is to illustrate the method used for the spatial analysis of precipitation. The selection of the case study (pag 11, lines 1-3) is based on the fact that for that specific day almost all the gauges have measured precipitation. This way, we thought the Figures would have been better suited for showing the step-by-step reconstruction of the precipitation field. We have not considered any measure of goodness-of-fit in the selection of the case study.

We avoid giving a misleading impression about the quality of the spatial analysis by verifying the performance of our method and discussing its pros and cons (Sections 5 and 6). In particular, for RR we have found

that the station density and the terrain complexity are the main factors determining the quality of our results, while the season of the year is a less important factor. These findings are in line with previous findings (e.g. Hofstra, 2008) and we have reported them in the paper (Discussion, page 16 lines 21-22; Conclusions, page 17 line 31).

For the reasons mentioned above, we believe that by adding another example application for precipitation we will not add significant information. An example of the performance of our method for a convective episode is reported in the following (see Section "Figures" below). At the beginning of August 2019 a thunderstorm occurred over Oslo (https://www.nrk.no/norge/se-hvordan-styrtregnet-splittet-oslo-1.14649700). This is an example of local-scale convective episode. Figure 4+ is equivalent to Figure 4 in the manuscript. Figure 5+ is equivalent to Figure 5 in the manuscript. Figure 6+ is equivalent to Figure 6 in the manuscript. The color scales are the same as in the manuscript. The key-messages one can extract from the Figures+ are rather similar to the ones that are described in the manuscript.

However, we think the Reviewer raises an interesting and important point. The readers may be interested in having a look at some precipitation (or temperature) fields before downloading the whole seNorge_2018 dataset. For this reason, we have included in the text some links to websites where it is possible to plot seNorge_2018 fields (see the Section "Code and data availability").

*4. Why is the section 5.3 analysis limited to winter? Surely it would be valuable to show this for at least summer in addition, if not all seasons to build user confidence and understanding of the product strengths and limitations?*

**Reply:** In the case of temperature, the occurrence of large errors as a function of station density for the summer season is discussed in Section 5.1.1 and shown in the bottom row of Fig. 7. Section 5.3 discuss more in detail the same issue but only for wintertime temperatures, such that the

limitations of our methods are emphasized. We have modified a bit the text to make this point clear to the reader.

In the case of precipitation, the verification shows that the season of the year is a less important factor than the station density (see the answer to Point 3). For this reason and in order to achieve more robust estimates, we have considered the whole dataset in Sec. 5.3.

We recognize there is a mistake in the text (page 15. Line 7). We have modified the text to correct the mistake.

*5. Generally the figure captions are too short. Often text should be moved from the main body to the caption. The figure captions need to provide all information necessary for a reader to understand and interpret the figure and oftentimes this is missing.*

**Reply:** we have revised all the figure captions.

*Minor comments*

*1. Given the caveats rightly stated in the discussion the abstract at lines 11-12 on page 1 seems a bit unduly definitive?*

**Reply:** In lines 11-12, two characteristics of the interpolation procedure are reported, such as: (i) the aggregated temperatures are interpolated separately (ii) physical consistency among them is enforced. Both statements are true.

*2. P.3 line 21 described and discusses -> described and discussed*

**Reply:** we have modified the text as suggested.

*3. P.4 line 13 so to -> so as to*

**Reply:** we have modified the text as suggested.

*4. P.4 line 20 as precipitation -> as the precipitation*

**Reply:** we have modified the text as suggested.

*5. P.6 line 14 tend -> Tends?*

**Reply:** we have modified the text "In the vicinity of an observation the IDI field is approximately equal to 1…"

*6. Much of the paragraph starting p.6 line 28 should be moved to the figure caption rather than the text. Also: IDI equals to -> IDI equal to*

**Reply:** we have modified the text as suggested.

*7. I would provide a very brief synopsis of the section 6 discussion at p.7 line 15-16*

**Reply:** The cross-checking is an important part and it is briefly summarized in the first paragraph of Sec. 3.1. We do believe that this paragraph is a brief synopsis of Sec. 6.

*8. P.9 line 31 so to transform -> either to transform OR so as to transform*

**Reply:** we have modified the text as suggested.

*9. P.13 lines. 18-20 should be in an expanded figure caption instead of the main text.*

**Reply:** we have modified the text to avoid repetition with the figure caption.

*10. Please rephrase p.13 line 24 – it should not be about belief.*

**Reply:** we have modified the text as suggested.

*11. P.15 line 4 In this paragraph -> Next*

**Reply:** we have modified the text as suggested.

*12. Figure 3 caption projection of the differences is, I assume, an error, but I'm not sure what you actually intend to state here.*

**Reply:** yes, it was an error. The caption has been corrected.

*13. Figure 4 caption. Is the colour bar dimensionless units? If so state so. If not then give the units. Regardless please clarify the caption.*

**Reply:** Yes, relative anomalies have dimensionless units and we have modified the caption to clarify it.

*14. Figure 5 caption. Please help the reader out here. Are larger or smaller scales preferable? How does this scale partition between the methodological aspects and the event specific aspects? The event in question is a large cyclonic event which may have broad spatial scales. To what extent are the scales shown here a result of the event specific nature instead of the method? Intuitively the two must be intertwined and may be remedied by showing e.g. a summer convective event as suggested in major comments.*

**Reply:** we have modified the caption of Figure 5 and extended in the text those parts describing the critical scale. In particular, we have discussed the points raised by the Reviewer.

*15. Figure 8 caption should end: See Fig 7 caption for further details.*

**Reply:** we have modified the text as suggested.

*16. Figure 13 lower right panel RR -> RR DJF for consistency with remaining panels*

**Reply:** This panel refers to RR and for this variable we have considered the whole dataset and not only the winter season (see also our answers to Major Comments 3 and 4). The figure caption has been modified to clarify this point.

**Figures**

[Figure]

**Figure 4+**. 2019-08-04. See Figure 4 caption in the manuscript.

[Figure]

**Figure 5+.** 2019-08-04 RR. Critical scale. See Figure 5 caption in the manuscript.

[Figure]

**Figure 6+.** 2019-08-04 RR analysis field. See Figure 6 caption in the manuscript.

[revised manuscript text omitted]